

# Feasibility of robust estimates of ozone production rates using satellite observations

**Amir H. Souri[1,2]\*, Gonzalo González Abad[3], Glenn M. Wolfe[1], Tijl Verhoelst[4], Corinne Vigouroux[4], Gaia Pinardi[4], Steven Compernolle[4], Bavo Langerock[4], Bryan N. Duncan[1], Matthew S. Johnson[5]**

[1]Atmospheric Chemistry and Dynamics Laboratory, NASA Goddard Space Flight Center, Greenbelt, MD, USA

[2]GESTAR II, Morgan State University, Baltimore, MD, USA

[3]Atomic and Molecular Physics (AMP) Division, Center for Astrophysics | Harvard & Smithsonian, Cambridge, MA, USA

[4]Royal Belgian Institute for Space Aeronomy (BIRA-IASB), Ringlaan 3, 1180 Uccle, Belgium

[5]Earth Science Division, NASA Ames Research Center, Moffett Field, CA, USA

\* Corresponding author: a.souri@nasa.gov

**Abstract.**

Ozone pollution is secondarily produced through a complex, non-linear chemical process. Our understanding of the spatiotemporal variations in photochemically produced ozone (i.e., $PO_3$) is limited to sparse aircraft campaigns and chemical transport models, which often carry significant biases. Hence, we present a novel satellite-derived $PO_3$ product informed by bias-corrected TROPOMI HCHO, $NO_2$, surface albedo data, and various models. These data are integrated into a parameterization that relies on HCHO, $NO_2$, HCHO/$NO_2$, $jNO_2$, and $jO^1D$. Despite its simplicity, it can reproduce ~90% of the variance in observationally constrained $PO_3$ with minimal biases in moderately to highly polluted regions. We map $PO_3$ across various regions in July 2019 at a 0.1°×0.1° spatial resolution, revealing accelerated values (>8 ppbv/hr) in numerous cities throughout Asia and the Middle East, resulting from the elevated ozone precursors and enhanced photochemistry. In Europe and the United States, such high levels are only detected over Benelux, Los Angeles, and New York City. $PO_3$ maxima are seen in various seasons, attributed to changes in photolysis rates, non-linear ozone chemistry, and fluctuations in HCHO and $NO_2$. Satellite errors result in moderate errors (40-60%) of $PO_3$ estimates over cities on a monthly average, while these errors exceed 100% in clean areas and under low light conditions. Using the current algorithm, we have demonstrated that satellite data can provide valuable information for robust $PO_3$ estimation. This capability expands future research through the application of data to address significant scientific questions about the locally-produced $PO_3$ hotspots, seasonality, and long-term trends.

## 1. Introduction

Tropospheric ozone ($O_3$) is a secondary pollutant formed through complex photochemical reactions involving various precursors, including nitrogen oxides ($NO_x$ = NO + $NO_2$), volatile organic compounds (VOCs), aerosols, and halogens (Kleinman et al., 2002, Simpson et al., 2015; Li et al., 2019). Ozone not only poses significant risks to human health and agricultural productivity but also influences the radiation



budget, thereby affecting the climate. To mitigate the problem of elevated locally-produced ozone, it is crucial to understand the spatiotemporal variability in ozone production rates ($PO_3$), defined as the number of ozone molecules generated through secondary chemical pathways in the atmosphere. Comprehensive studies of ozone chemistry, informed by observations, are typically confined to observationally-rich air quality campaigns (e.g., Cazorla et al., 2012; Ren et al., 2013; Mazzuca et al.; 2016; Souri et al., 2020a; Schroeder et al., 2020; Brune et al., 2022; Wolfe et al., 2022; Souri et al., 2023), which are sparse in time and space.

Significant advancements have been achieved in using various measurable ozone indicators to simplify the non-linear relationship between $PO_3$ and $NO_x$ and VOCs into linear forms (Sillman and He, 2002). These forms include $NO_x$-sensitive (where $PO_3$ is sensitive to $NO_x$), VOC-sensitive (where $PO_3$ is sensitive to VOCs), and the transitional regimes (where $PO_3$ is sensitive to both $NO_x$ and VOCs). Among the numerous proposed indicators, the ratio of formaldehyde (HCHO) to nitrogen dioxide ($NO_2$) (known as FNR) has gained popularity (Tonnesen and Dennis, 2000a,b), despite its less effective performance compared to the $H_2O_2/HNO_3$ ratio in fully explaining the $HO_x$-$RO_x$ cycle (Silman and He, 2002; Souri et al., 2023). The preference for FNR stems from the fact that both quantities can be informed by UV-Vis radiance data, such as those provided by the Ozone Monitoring Instrument (OMI) and the TROPOspheric Monitoring Instrument (TROPOMI) (Martin et al., 2005; Duncan et al., 2010; Choi et al., 2012; Choi and Souri, 2015a, b; Jin and Holloway, 2015; Jin et al., 2017; Schroeder et al., 2017; Souri et al., 2017; Jeon et al., 2018; Tao et al., 2022). Several limitations associated with the application of satellite-based FNR have been identified such as i) the inherent limitation of understanding the radical termination in the $RO_x$-$HO_x$ cycle (Souri et al., 2020a; Souri et al., 2023), ii) the challenges associated with converting the column vertical density to the near-surface concentrations (Jin et al., 2017; Schroeder et al., 2017; Souri et al., 2023), iii) spatial representativity associated with large satellite pixels (Souri et al., 2020a, 2023; Johnson et al., 2023), and iv) the retrieval errors (Souri et al., 2023; Johnson et al., 2023). Souri et al. (2023) concluded that the retrieval errors make up the largest portion of total errors associated with FNR. These errors are becoming smaller with better sensor designs, retrieval algorithms, and calibration over time.

While the characterization of ozone regimes offers valuable insights for regulators to prioritize effective emission control strategies, it does not provide information about the magnitude of $PO_3$ or the absolute quantities of $PO_3$ derivatives relative to its precursors. Consequently, chemical transport models under various emission scenarios are typically employed (e.g., Pan et al., 2019). These models allow for the execution of process-based scenarios to elucidate the response of $PO_3$ to different emissions and can simulate four-dimensional $PO_3$ data. However, the results of these simulations are based on various assumptions and inputs, which carry significant uncertainties. Therefore, it is essential to optimize some of the models' prognostic inputs using observations through inverse modeling/data assimilation. The primary advantage of inverse modeling/data assimilation using satellite observations is its ability to account for satellite errors and eliminate the influence of the a priori profile, thereby carrying only radiance information into the emission estimation. Numerous studies have utilized satellite observations to constrain $NO_x$ and VOC emissions for various applications (e.g., Stavrakou et al., 2016; Souri et al., 2016; Miyazaki et al., 2017; Souri et al., 2017; Souri et al., 2020b; Souri et al., 2021; Choi et al., 2022; DiMaria et al., 2023). Souri et al. (2020b) made an early attempt to simultaneously optimize both $NO_x$ and VOC emissions over East Asia for a more accurate representation of $PO_3$. Their joint-inversion was able to account for the intertwined relationship between HCHO-$NO_x$ and $NO_2$-VOC. However, the execution of chemical transport models optimized by multiple satellite observations remains prohibitively expensive, particularly for high-resolution domains demanded by regulatory agencies.

Data-driven methods for estimating $PO_3$ have emerged as a more cost-effective alternative to physics-based methods. While using constrained chemical transport models provides a relatively robust framework grounded in some explicit governing equations, they require extensive computation resources and expertise. Conversely, data-driven algorithms make use of large datasets to identify patterns and make predictions with much reduced computational expenses. However, it is important to recognize that data-



driven algorithms lack the ability to provide solid physical interpretability and generalizability. Despite this fundamental limitation, they are sensible tools for applications where rapid analysis over a wide spatial coverage is prioritized. Data-driven parameterizations for several components of atmospheric chemistry such as OH (Anderson et al., 2022) and dry deposition (Silva et al., 2019) have been crafted for this reason. However, to our best knowledge, Chatfield et al. (2010) and Souri et al. (2023) are the only studies that attempted to empirically parameterize $PO_3$ using the information of HCHO and $NO_2$ mixing ratios. Inspired by those works, we developed a novel product using TROPOMI observations in conjunction with atmospheric models to provide $PO_3$ and associated errors within the planetary boundary layer (PBL) across the globe. This enabled us to map $PO_3$ across various regions at fine scales (i.e., 0.1×0.1 degrees) for the first time.

## 2. Data

### 2.1. Aircraft

To study $PO_3$, we use various aircraft observations from several NASA and NOAA atmospheric composition campaigns. We have selected three sets of aircraft campaigns for the purpose of $PO_3$ estimation, targeting: i) urban/suburban air quality, including Deriving Information on Surface Conditions from Column and Vertically Resolved Observations Relevant to Air Quality (DISCOVER-AQ) Baltimore-Washington (2011), DISCOVER-AQ Houston-Texas (2013), DISCOVER-AQ Colorado (2014), and the Korea United States Air Quality Study (KORUS-AQ) (2016) (Crawford et al., 2021); ii) remote areas including Atmospheric Tomography Mission (ATOM) (Thompson et al., 2022) and Intercontinental Chemical Transport Experiment (INTEX) phase B (Singh et al., 2009); iii) a mixture of isoprene-rich environment and large emitters, including SENEX (Southeast Nexus) (Warneke et al., 2016). Figure 1 shows the location of these campaigns. Inspired by the study of Miller and Brune (2022), we list their "when, where, why" characteristics in Table S1.

For aircraft campaigns targeting polluted areas, including DISCOVERs, KORUS-AQ, SENEX, and SEAC4RS, we use 10-sec merged data, whereas, for other measurements taken in relatively remote areas, such as INTEX-B and ATOMs, we used 30-sec merged data. A more detailed description of the measurements is provided in Section 3.2. We exclude times with no measurements of NO, $NO_2$, or HCHO. The concentrations of OH and $HO_2$ were only measured during INTEX-B, ATOMs, and KORUS-AQ. Likewise, we void any data points lacking either $HO_2$ or OH measurements. There are frequent gaps in some measurements, especially for VOCs, because of instrument issues or measurement techniques. Following Souri et al. (2020a), Miller and Brune (2020), Souri et al. (2023), and Bottorff et al. (2023), we fill the gaps in measurements using a linear interpolation method with no extrapolation allowed beyond 15 minutes. We drop any remaining gaps from the analysis. To better capture the rapid fluctuation of VOCs, we pick the PTR-TOF-MS instrument with high temporal resolution over the whole air sampler (WAS) when both instruments have measured the same quantity. Regarding the INTEX-B campaign, we drop isoprene observation due to infrequent samples downgrading the performance of our box model.



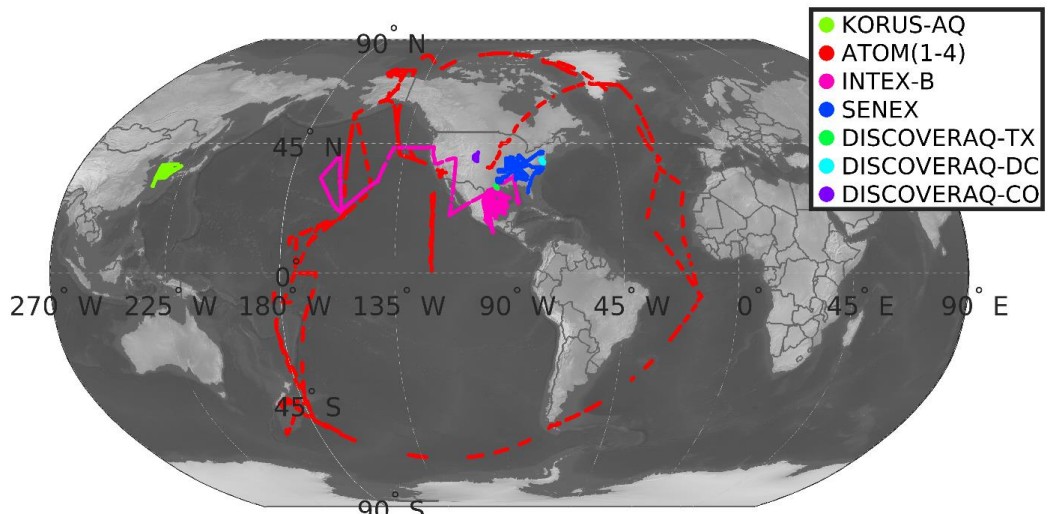


**Figure 1.** The location of seven different atmospheric composition aircraft campaigns used in this study.
### 2.2. TROPOMI NO₂ and HCHO
We use the recently reprocessed daily level-2 (L2) TROPOMI tropospheric NO$_2$ and total HCHO
columns derived from UV-visible radiances onboard the European Space Agency's (ESA's) Sentinel-5
Precursor (S5P) spacecraft (~328-496 nm) (Veefkind et al., 2012, De Smedt et al. 2021; van Geffen et al.,
2022). This sensor has been operational since May 2018, providing global coverage of NO$_2$ and HCHO at
~1:30 local standard time at the Equator. Since NO$_2$ and HCHO are optically thin absorbers in the UV-
Visible, meaning their concentrations do not substantially affect the sensitivity of the radiance to the optical
thickness of the absorber, the retrieval follows the conventional two-step algorithm involving spectral fitting
for Slant Column Density (SCD) retrieval and Air Mass Factor (AMF) calculations for SCD to Vertical
Column Density (VCD) conversion. The product has a spatial resolution of 7.2 km (5.6 km as of August
2019) by 3.6 km at nadir. To remove unfit measurements, we use the provided quality flag (*q_value*) and
choose only those above 0.75 for NO$_2$ and 0.5 for HCHO. As the L2 product does not come in a regular
grid, we use a mass-conserved regridding technique based on barycentric linear interpolation to map out
the data onto a 0.1°×0.1° regular grid.
van Geffen et al. (2022) demonstrated that the reprocessed TROPOMI tropospheric NO$_2$ columns
exhibit a good level of correspondence with those obtained from ground-based MAX-DOAS sky
spectrometers, with a correlation of 0.88 and a median bias of -23%, improving on the older product
versions which were biased low by about 30% with respect to ground-based measurements at polluted sites
(Verhoelst et al., 2021). More information about new modifications and their impacts on the retrieval can
be found in van Geffen et al. (2022).
The studies of Vigouroux et al. (2020) and De Smedt et al. (2021) validated the reprocessed
monthly-mean TROPOMI HCHO columns against FTIR and MAX-DOAS observations and found a good
correlation above 0.8 with a negative bias of 20-30% for polluted sites. The bias tends to be slightly positive
or neutral over clean sites.
### 2.2.1. Error characterization of TROPOMI NO₂ and HCHO using sky-radiance retrievals
To propagate TROPOMI retrieval errors to the PO$_3$ product and to remove potential biases, we
assume three origins for errors: i) random errors resulting from instrument noise, ii) a fixed additive



component that is magnitude-independent, and iii) unresolved systematic biases that are multiplicative and irreducible by oversampling. The first component is derived from the column precision variable provided along with the L2 product. In the spatial domain, we interpolate the squares of this error the same of way we map the irregular L2 pixels into the 0.1°×0.1° regular grid. Moreover, to mitigate this error, its squares are averaged for over a month. Two other errors are determined by comparing FTIR (for HCHO) and MAX-DOAS (for tropospheric $NO_2$) with TROPOMI data (Section 4.3.3). Detailed explanation of how these datasets are paired can be found in Vigouroux et al. (2020) and Verhoelst et al. (2021). Both datasets cover the period of 2018-2023.

To achieve an optimal linear fit ($y = ax + b + \varepsilon$) between the paired observations, we follow a Monte-Carlo Chi-squares minimization such that $\chi^2 = \sum \frac{[y - f(x_i, a, b)]^2}{\sigma_y^2 + a^2 \sigma_x^2}$ is minimized. $\sigma_y^2$ and $\sigma_x^2$ are the variances of $y$ (TROPOMI) and $x$ (the benchmark), respectively. In terms of TROPOMI $NO_2$ and HCHO, the errors are populated based on the L2 information. According to Verhoelst et al. (2021), a fixed error of 30% is assumed for MAX-DOAS $NO_2$ observations whose values are above 1.4 Pmolec/cm$^2$. Because of the detection limit of MAX-DOAS $NO_2$, we set errors for values below that threshold to 1.4 Pmolec/cm$^2$. The FTIR retrieval errors described in Vigouroux et al. (2020) were used to populate the errors associated with this benchmark. The minimization is performed 10000 times, each with a set of random perturbations of $x$ and $y$ within their respective prescribed errors. This approach allows us to assess the robustness of the estimates across the range of errors associated with each data point.

The offset (a uniform additive term) and the slope (multiplicative error) drawn from the ground validation are used to correct the biases associated with TROPOMI via:

$$VCD_{bias-corrected} = \frac{VCD_{original} - offset}{slope} \tag{1}$$

Since there are errors associated with this adjustment resulting from instrument and representation errors, we augment errors of the slope and offset to the total error and label them constant errors ($e_{const}$) via:

$$e_{const}^2 = e_{offset}^2 + e_{slope}^2 \times VCD_{bias-corrected}^2 \tag{2}$$

where $e_{offset}$ and $e_{slope}$ are errors of offset and slope calculated from the linear regression. Ultimately, the sum of all three errors constitutes the total errors given:

$$e^2 = e_{const}^2 + \frac{1}{m^2} \sum_{i=1}^{m} e_{random,i}^2 \tag{3}$$

where $m$ is the number of samples for a given grid and timeframe and $e_{random}$ is random errors.

### 2.3. TROPOMI Surface Albedo

To account for the effect of surface albedo on photolysis rates (Section 2.5), we use a newly developed algorithm based on the directionally dependent Lambertian-equivalent reflectivity (DLER) UV surface albedo climatology made from TROPOMI radiance (Tilstra et al., 2024). This new database leverages 60 months of TROPOMI reprocessed radiance and is produced at the grid resolution of 0.125°×0.125°. The product has outperformed traditional LER products such as OMI when both were compared to MODIS surface BRDF results (Tilstra et al., 2024).

### 2.4. MERRA2-GMI

To convert vertical column densities of HCHO and $NO_2$ from TROPOMI to their volume mixing ratios in the PBL region, we use the MERRA2-GMI model (https://acd-



ext.gsfc.nasa.gov/Projects/GEOSCCM/MERRA2GMI/, last access: 10 Sep 2023). This model is a NASA's
Goddard Earth Observing System (GEOS) Chemistry-Climate Model (CCM) run spanning for the period
of 1980-2019, exploiting MERRA2 (Modern Era Retrospective analysis for Research and Applications) to
constrain meteorological fields (Orbe et al., 2017). The model uses the Global Modeling Initiative (GMI)
chemical mechanism (Duncan et al., 2007; Strahan et al., 2007), which involves over 120 species and 400
reactions. It has a resolution of approximately 0.625° longitude by 0.5° latitude with 72 vertical layers
stretching from the surface up to 0.1 hPa. Additional information about the configuration of this model can
be found in Strode et al. (2019).
*2.5. TUV NCAR Photolysis Rates Look-up Table*
To estimate photolysis rates of $JNO_2$ ($NO_2+hv$) and $JO^1D$ ($O_3+hv$), we use a comprehensive look-
up table provided by the F0AM model (Section 3.2) created for clear-sky conditions. This look-up table is
based on the calculation of more than 20,064 solar spectra over a wide range of SZA (0:5:90°), altitude
(0:1:15 km), overhead total ozone column (100:50:600 DU), and surface UV albedo (0:0.2:1) using
NCAR's Tropospheric Ultraviolent and Visible radiation model (TUV v5.2) and cross sections and quantum
yields from IUPAC and JPL (Wolfe et al., 2016). The L2 TROPOMI granule information populates SZA,
surface elevation, and surface UV albedo, while overhead total ozone columns are obtained from MERRA2-
GMI (Section 2.4) which is found to agree well with satellite observations (Souri et al., 2024). Any values
between these tables are bilinearly interpolated for a smoother result.

**3. Methods**

*3.1. LASSO*
Through the use of multi-linear regression models, it is possible to establish a simple but robust
relationship between multiple variables and a target. However, when dealing with a large number of
variables, there is a chance of introducing overfitting issues. This can lead to predictions that are either
overly optimistic or unrealistic for values outside of the training dataset. To avoid this, it is recommended
to simplify the model by removing variables that are loosely connected with the target or highly correlated
with others. This process is known as "model shrinkage" and can narrow down the number of possible
solutions (i.e., variance) at the cost of increasing the biases between the observed target and predictions.
Ideally, we want a model that minimizes the sum of the bias and the variance. To achieve this, we can use
LASSO (least absolute shrinkage and selection operator) (Tibshirani, 1996) consider a regression,

$$Y = X\beta + \alpha + \varepsilon \tag{4}$$

with response $Y = (y_1, ..., y_n)^T$, $n \times p$ explanatory variables $X$, coefficients $\beta = (\beta_1, ..., \beta_p)^T$, an intercept $\alpha$,
and noise variables $\varepsilon = (\varepsilon_1, ..., \varepsilon_n)^T$. We can label the regression model sparse when many of $\beta$ values are
zero, and we can label it high dimensional when $p \gg n$. LASSO attempts to select variables such that the
following cost function is minimized:

$$(\hat{\alpha}, \hat{\beta}) = argmin \left\{ \|Y - X\beta - \alpha\|_2 + \lambda \sum_{i=1}^{p} |\beta_i| \right\} \tag{5}$$

The first term on the right side of Eq.5 minimizes the squares of the residuals, whereas the second term
reduces the sum of absolute value of coefficients resulting in a simpler model with fewer parameters.
Without the second term, the regression model becomes an ordinary least-squares estimation. The most
critical element here is $\lambda$, a non-negative regularization factor. A large $\lambda$ results in more aggressive
regularization leading to more model shrinkage, whereas a small value preserves a high dimensional model.
To optimize this value, we discretize $\lambda$ in 100 values between $10^{-4}$ up to $10^1$, divide the training dataset into



10 folds, determine the average of cross-validated error prediction among all folds, and find $\lambda$ that yields
the smallest error. The final solution ensures a balanced model with respect to model parsimony and bias.
All explanatory variables are standardized during the regularization procedure such that their mean becomes
zero and their standard deviation one.

### 3.2. Photochemical box modeling

To produce training data sets for LASSO-based $PO_3$ estimation, we use the Framework for 0-D
Atmospheric Modeling (F0AM) v4 box model (Wolfe et al., 2016), constrained by a wide range of
observations. These observations ensure that the model achieves a realistic range of values found in the
atmosphere. We follow past setups which apply the Carbon Bond 6 (CB06, r2) chemical mechanism in
F0AM (Souri et al., 2020a; Souri et al., 2023). The model is constrained by aircraft data, including
meteorology, photolysis rates, and trace gas concentrations. The model configuration and observations used
are listed in Table S2.
Once the model is initialized and held constant with respect to a wide range of constraining
quantities, it runs at 30 minutes integration time cycling for five days to approach a steady-state
environment. Several key compounds including OH, $HO_2$, HCHO, PAN, NO, and $NO_2$ are initialized with
aircraft observations but they are left free to cycle with incoming solar radiation variability. These
compounds play a crucial role in validating the efficacy of model performance as well as the adequacy of
observations used as constraints. In particular, allowing HCHO to vary freely enables us to assess whether
our mechanism for VOC treatment, steady-state, and the number of measured VOCs suffice to reproduce
its concentrations reasonably. Although the individual concentration of $NO_2$ and NO are not constrained,
we constrain total $NO_x$ ($NO+NO_2$). Not all aircraft campaigns measured all photolysis rates included in the
chemical mechanism. We first initialize the photolysis rates included in CB06 using the look-up-tables
described in Section 2.5. If any photolysis reaction rates in CB06 were measured, we replace the initial
guess with the observed values. For those reactions with photolysis rates not been measured, we apply a
scaling factor made of the average of the ratio of the observed J-values to the modeled J-values. This
approach is a sensible choice for accounting for large particles such as clouds, as their extinction coefficient
is somewhat non-selective in the UV-Vis range; however, applying a wavelength-independent scaling factor
may introduce some biases for optically complex environments introduced by aerosols.
It is essential to acknowledge the inherent limitations of a box model in our research. The model
does not consider the diverse physical loss pathways that trace gases may undergo, including deposition
and transport. As a result, we have simplified the physical loss by employing a first-order dilution rate set
to $1/86400$ $s^{-1}$, equivalent to a lifetime of 24 hours. This approach ensures that unconstrained trace gases
that take longer to break down do not accumulate over time. Exact knowledge of dilution factors requires
knowing molecular and turbulent diffusion, entrainment and detrainment, and deposition rates, all of which
are unknown at the micro-scale level of aircraft observations. Nonetheless, studies of Brune et al. (2022)
and Souri et al. (2023) showed that $HO_2$, OH, $NO_x$, and HCHO are relatively immune to the choice of the
dilution factor, whereas $RO_2$ mixing ratios can depart introducing some biases in $PO_3$ estimates.
We determine simulated $PO_3$ by:

$$PO_3 = FO_3 - LO_3 \qquad (6)$$

where $LO_3$ is all possible chemical loss pathways of ozone (negative stoichiometric multiplier matrix) and
$FO_3$ is all possible chemical pathways producing ozone molecules (positive stoichiometric multiplier
matrix). This calculation is theoretically equivalent to a value obtained from a chemical solver quantifying
the number of ozone molecules produced/lost for each model timestep. The adoption of Eq.3 facilitates the
direct comparison of $PO_3$ estimations with those derived from other models, including CTM-based results
(see Figure 10 in Souri et al., 2021). Furthermore, it allows for a seamless integration of these estimates
into Lagrangian transport models for ozone forecasting purposes.




### 3.3. Clustering

Using a classifier to group the large quantity and types of aircraft data into similar features allows us to study the primary contributors to $PO_3$ under different chemical, solar, and meteorological conditions. To accomplish this, we employ a widely-used technique known as $k$-means, which has been used in a variety of applications (e.g., Beddows et al., 2009; Souri et al., 2016b; Govender and Sivakumar, 2020). In this approach, centroids are distributed randomly throughout a multi-dimensional dataset, with each centroid representing a distinct class. The algorithm proceeds to assign a label to each data point by identifying its closest Euclidean distance to the centroids. Following the labeling of all data points, the algorithm updates the centroids based on the means of the newly-labeled group. This process continues iteratively until there is minimal change in the location of the centroids. It is worth noting that $k$-means does not guarantee an optimal solution, so we reinitialize the classification 1000 times with a new set of initial centroids. We select the result with the lowest value for the sum of the Euclidean distance among data points and centroids to ensure the outcomes are not influenced by random seeding.

Redundant features in the input can significantly compromise the effectiveness of the classification, so we apply principal component analysis (PCA) to the matrix of datasets ($Z$) with $n$ data points and $p$ features to reduce the dimension to a PCA-transformed matrix of $Z$ ($Z$) with the dimension $n \times q$, where q<p. Despite this reduction in dimension, $Z$ preserves a significant variance in $Z$, helping us to overcome the issues of dimensionality or overfitting.

We select 11 features simulated by the F0AM model, many of which are set to the observed values, or their precursors are observationally-constrained. These features include: SZA, $HCHO/NO_2$, $HCHO \times NO_2$, HCHO, $NO_2$, pressure, temperature, $jNO_2$, $jO^1D$, $H_2O$, and $NO_2/NO_y$ ($NO_y=NO+NO_2+PAN+HNO_3+$alkyl nitrate $+N_2O_5$). There are indeed correlations among these features such as SZA and $jNO_2$, or HCHO and $HCHO \times NO_2$; nonetheless, we have used PCA to eliminate the possibility of these correlated factors causing overfitting issues.

### 3.4. The estimation of $PO_3$

In order to predict $PO_3$, we have developed empirical equations using LASSO to link $PO_3$ with various relevant prognostic candidates related to ozone chemistry. A schematic presenting how this estimation can be done to provide daily $PO_3$ maps at the TROPOMI revisit time across the globe is shown in Figure 2. It is important to note that relying solely on linear regressions for a non-linear problem is not a viable approach. To address this, we have divided the data points into four distinct groups based on FNR values, meaning we divide a non-linear realm into smaller linear segments (i.e., an empirical linearization). In a study by Souri et al. (2023), a wide range of aircraft observations and box model results were used to determine that FNR~1.7 was a universal threshold for separating NOx-sensitive from VOC-sensitive regimes. We have found that by breaking down the datapoints into slightly weaker or stronger variations of the regimes, we can improve the accuracy of our results. As a result, we have established four distinct groups: VOC-sensitive (FNR<1.5), transitions (1.5<FNR<2.5 and 2.5<FNR<3.5), and NOx-sensitive (FNR>3.5). The coefficients and intercepts based on the LASSO regressions for each group were computed separately. From a long list of explanatory parameters, we selected SZA, temperature, pressure, $H_2O$, $jNO_2$, $jO^1D$, HCHO, and $NO_2$ as the most sensible candidates. The reasoning behind this selection will be discussed in Section 4.2.

Once the LASSO parameters are determined, we applied the linear functions to variables modeled/observed in the PBL region. We show that the LASSO method votes for dropping SZA, temperature, and pressure as they do not provide significant information on $PO_3$ compared to the rest. As for $jNO_2$ and $jO^1D$, we use the TUV NCAR's LUT described in Section 2.5. HCHO and $NO_2$ are based on converted the bias-corrected TROPOMI VCDs into PBL mixing ratios using MERRA2-GMI described in Section 2.4. To carry out the conversion, we multiply the satellite VCDs by the ratio of averaged modeled





mixing ratios of a target gas (i.e., $NO_2$ or HCHO) in the PBL region divided by modeled VCDs. The PBL
field also comes from MERRA2-GMI.

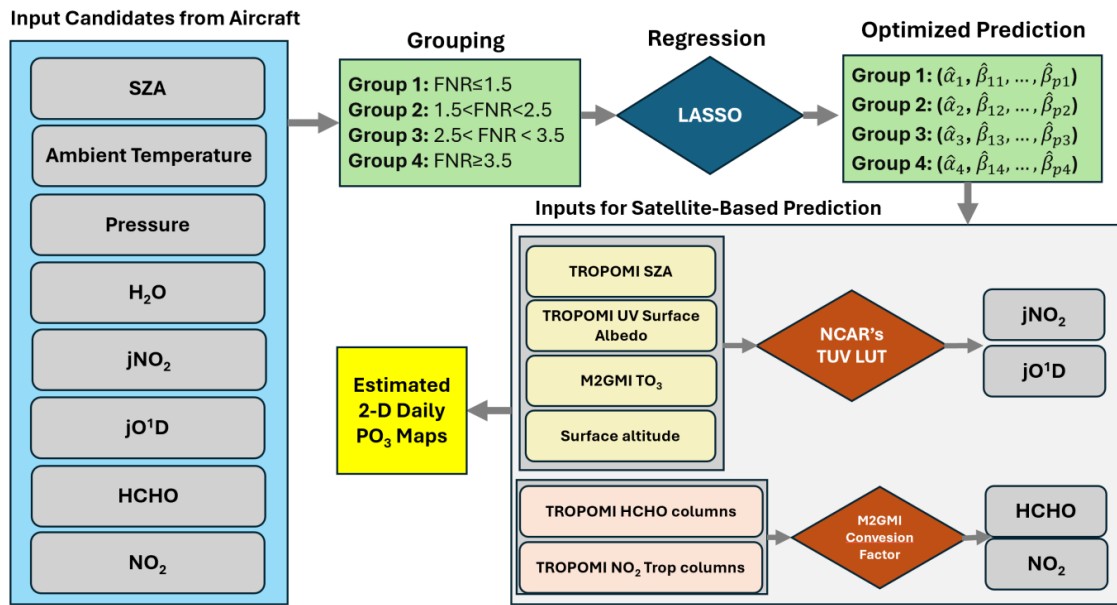

**Figure 2.** Schematic illustration of daily $PO_3$ estimation calculated in this study. This process consists of
two major steps: formulating $PO_3$ as a function of various prognostic inputs derived from the box model
results, and predicting $PO_3$ based on optimized features/coefficients suggested by LASSO and using
information obtained from TROPOMI, TUV, and M2GMI.

## 4. Results and Discussion

### 4.1. Box Model Validation

In order to assess the accuracy of the assumptions used in the box model's setup, which involves
factors such as chemical mechanism, dilution rate, and photolysis rate correction, we will compare the
simulated values of HCHO, $NO_2$, NO, PAN, $HO_2$, and OH with their actual measured values. This
comparison will help us determine if our model falls within an acceptable range of errors as seen in other
reputable photochemical box modeling studies. This comparison is represented in Figure 3, which displays
a scatterplot of the data collected from all seven aircraft campaigns. A discussion on each parameter follows:
HCHO – The box model is proficient in capturing over 77% of variance in observations with less
than 15% absolute bias. While many box modeling studies prefer to have this compound constrained to
potentially enhance the representation of HOx, it comes with the trade-off of hindering us from validating
the number/quality of observed HCHO precursors and/or the VOC treatment. Besides the study of Souri et
al. (2023), Marvin et al. (2017) is one of the few studies that did not constrain this compound to verify the
efficacy of different pathways involved in HCHO formation and loss simulated by various chemical
mechanisms. Marvin et al. (2017) reproduced HCHO formation during the SENEX campaign using the
CB06 mechanism with a $R^2$=0.66 and a bias of 32% at 1-min averaged samples. Compared to that study,
we recreate 86% variance in observed HCHO during the same campaign with a bias of 23% (Figure S1) at
10-sec averaged samples. The remaining unresolved variance can be attributed to an incomplete list of VOC
measurements for several campaigns including DISCOVER-AQs and errors of VOCs measurements. It is



unlikely for the chemical mechanism to be reason for this, as Marvin et al. (2017) did not observe substantial
differences in $R^2$ values among various chemical mechanisms including the near-explicit MCM. A mild
underestimation of HCHO could be likely due to the steady-state assumption, fixed arbitrary dilution factor,
or uncertain isoprene chemistry (Archibald et al., 2000; Wolfe et al., 2016; Marvin et al., 2017).
$NO_2$ and NO – Comparisons for both species demonstrate a high degree of correspondence for
values above 0.1 ppbv. Nonetheless, we have noted a substantial amount of fluctuation in the simulations
in clean regions, particularly for NO. While we cannot rule out the possibility of chemical mechanism
uncertainty contributing to this deviation, the reported measurement errors for $NO_2$ and NO are usually
±0.05 ppbv and ±0.1 ppbv, respectively. Consequently, it is likely that the measurements error resulted in
more spread in comparison.
PAN – Our model reproduced 61% of the variance observed in PAN with a marginal absolute bias.
According to Xu et al. (2021), the presence of oxygenated VOCs, particularly acetaldehyde, and the
$NO/NO_2$ ratio are key factors controlling PAN levels. While we have constrained acetaldehyde, variations
in the $NO/NO_2$ ratio in heavily polluted regions (where $NO_x$ levels exceed 1 ppbv) could potentially lead
to biases in PAN simulations. Furthermore, our model's dilution factor has been arbitrarily set, and it is
possible that any bias caused by this factor has been canceled out by other effects, leading to seemingly
bias-free performance. However, Souri et al. (2023) showed that an incorrect dilution factor can
significantly impact PAN performance, causing a sharp decline in $R^2$ resulting in a value below 30%.
Therefore, the fact that our box model has performed well with respect to PAN could be an indication that
our choice of the dilution factor is not too unrealistic.
$HO_2$ and OH – Based on our analysis of $HO_2$ and OH simulations during KORUS-AQ, INTEX-B,
and ATOMs, we have found a reasonable level of correspondence ($R^2$>0.6) with the performance in
previous studies conducted by Souri et al. (2020), Brune et al. (2022), Miller and Brune (2022), and Souri
et al. (2023) that focused on some of these campaigns. Although the box model OH simulations reported in
Brune et al. (2019) during ATOMs seemed to be better than ours ($R^2$~0.8 vs $R^2$~0.6), it is important to
consider that their observations were averaged over 1-minute intervals as opposed to our 30-second
intervals. It should also be noted that there can be large errors in ATHOS $HO_x$ measurements of up to ±40%
(Miller et al., 2022), so recreating the exact variance in the observations should not be the main objective.
Nonetheless, the performance of our simulations in terms of $HO_x$ compared to observations suggests that
the number of measured compounds and chemical mechanisms used in the model was effective. Our
model's performance with respect to $HO_x$ is comparable to more sophisticated mechanisms that encompass
a larger number of measured species (Brune et al., 2022; Miller and Brune, 2022).
Overall, while there are inevitably some differences between the box model results and
observations, they are consistent with what other studies have found in similar aircraft campaigns. Our
extensive box model results, which consider a variety of meteorological, chemical, and photolysis rates,
demonstrate satisfactory results for unconstrained compounds across a wide range of atmospheric
conditions. This suggests that our training dataset from the box model is a reliable source for understanding
local $PO_3$.
It is important to note that even if a simulated data point does not match up perfectly with actual
observations, it still plays a role in establishing $PO_3$ and other explanatory variables. Hypothetically, one
can generate synthetic training data points by running the box model under random numbers for the inputs;
but only a fraction of those can be truly observed in nature. Therefore, a mild outlier in our training dataset
should be viewed as less likely to occur in nature (presuming that these campaigns could represent all
conditions happening in nature), but still a valuable data point drawn from a physical model that can be
used to bridge $PO_3$ with explanatory variables.

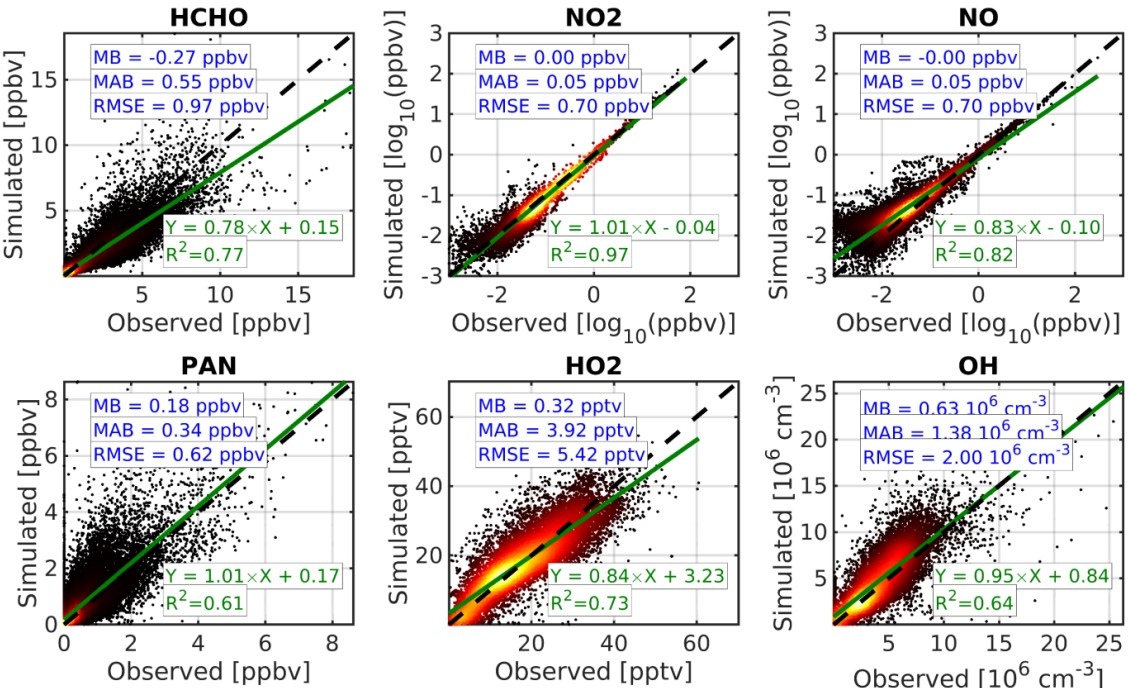

**Figure 3.** The scatterplot comparison of simulations with observed concentrations for six unconstrained species. More than ~133,000 observations are used for HCHO, $NO_2$, NO, and PAN. $HO_x$ data points are limited to ~55,000 observations. Heat maps show the density of the data. Linear fits are calculated using the least squares method. A high $R^2$ value for HCHO indicates reasonable VOC mechanism and measurements, given that some aircraft campaigns measured only a handful of VOCs. A relatively high $R^2$ with minimal bias for PAN implies that the choice of the dilution factor is not unrealistic. A good representation of $HO_x$ can indicate a reasonable prediction of the $HO_x$-$RO_x$ cycle controlling $PO_3$.

## *4.2. Classification of aircraft data*

Following the method described in Section 3.3, we cluster the cloud of aircraft data (~ 133k points) into seven distinct classes. We describe them using three categories: pollution level, altitude, and SZA. Figure 4 illustrates the violin plot of these classes for various chemical, solar, and meteorological conditions. Figure 5 shows their corresponding violin plot of simulated $PO_3$. A discussion of each class and their relationship to $PO_3$ follows:

C1 (clean, high altitude, high SZA) – Characterized by high altitude flights, cold ambient temperature, and negligible water vapor content, this class consists of observations that were typically taken during relatively high SZA with a median of 50°. While high altitude observations in clear-sky conditions often should have large photolysis rates due to reduced overhead ozone, the relatively high SZA of this class leads to low photolysis rates. FNRs tend to be large in this class due to a higher amount of HCHO over $NO_2$, and FNP (HCHO×$NO_2$) and $NO_2/NO_y$ ratios are low due to the pristine conditions. The lack of sufficient ozone precursors and reduced photochemistry make this class undergo the lowest $PO_3$ rates with a median of 0.11 ppbv/hr.

C2 (clean, high altitude, low SZA) - This category represents samples collected in low SZA conditions, resulting in the highest photolysis rates among all classes. The mass of ozone precursors and the ozone



sensitivity condition are similar to those in C1. However, C2 $PO_3$ rates are approximately 60% higher than
C1 due to increased photochemistry.
C3 (moderately clean, medium altitude, high SZA) - This class is characterized by observations collected
in mid-altitudes and high SZA. Airsheds in C3 experienced relatively more polluted air compared to C1
and C2 due to being closer to the surface. Photolysis rates are smaller than C1 possibly because of higher
ozone overhead, although we cannot rule out the varying surface albedo between the classes. Despite the
lower photolysis rates, C3 $PO_3$ (0.28 ppbv/hr) is larger than that of C2 and C1, indicating that pollution
levels can have a more significant impact than favorable conditions for photochemistry.
C4 (moderately clean, medium altitude, low SZA) - This category is distinct from C3 in terms of lower
SZA (resulting in more photochemistry) and a slightly smaller number of ozone precursors. As a result of
the lower ozone precursor concentration, not only is C4 $PO_3$ (0.19 ppbv/hr) lower than C3, but also is not
significantly different from C2. This again implies that the amount of ozone precursors is more important
than the photochemistry for these conditions.
C5 (extremely polluted, low altitude, low SZA) - This class features the highest amount of ozone precursors
(median FNP ~ 58 $ppbv^2$) among all classes. Furthermore, it is characterized by low photolysis rates due to
its proximity to the surface, and high $NO_2/NO_y$ indicative of localized polluted airshed. Unlike the previous
classes, this class has the lowest FNR, indicating that it is mainly located in the VOC-sensitive regime. C5
$PO_3$ values are much higher than the previous classes, with a value of 3.0 ppbv/hr.
C6 (polluted, low altitude, low SZA) - While this class shares similar features with C5 in terms of altitude,
photolysis rates, and meteorology, it experiences a lower FNP (median of 8 $ppbv^2$). Despite the lower FNP,
C6 has the highest amount of $PO_3$ (5.2 ppbv/hr) among all classes. This is a result of reduced non-linearities,
as this class does not often fall into an extreme VOC-sensitive regime (median FNR ~ 1.0) where nitrogen
oxides ($NO_x$) can hamper ozone production. This tendency coincides with Souri et al. (2023) which also
found that the highest amount of $PO_3$, lied between the transitional regimes, gravitated towards VOC-
sensitive because of abundant ozone precursors and reduced negative chemical feedback of $NO_x$.
C7 (moderately polluted, low altitude, high SZA) - C7 is characterized by aged air close to the surface with
slightly higher photolysis rates than C5 and C6. C7 $PO_3$ is 2.5 ppbv/hr, only slightly smaller than C5 despite
much lower FNP (median of 0.9 $ppbv^2$). This could be caused by the combined effect of higher photolysis
rates and reduced non-linear ozone chemistry.
The analysis of aircraft data has revealed that the levels of HCHO and $NO_2$, as well as the rates of
$jNO_2$ and $jO^1D$ photolysis, play an important role in influencing $PO_3$. Additionally, FNRs can offer insights
into the sensitivity of $PO_3$ to its main precursors. These findings align with numerous other studies that
have examined the factors driving $PO_3$ (e.g., Duncan and Chameides, 1998; Thornton et al., 2002; Kleiman
et al., 2002; Gerasopoulos et al., 2006; Chatfield et al., 2010; Baylon et al., 2018; Wang et al., 2020; Souri
et al., 2023). Consequently, our $PO_3$ estimates will incorporate HCHO, $NO_2$, $jNO_2$, $jO^1D$, and FNR. While
the cluster analysis did not definitively indicate whether meteorological conditions impact $PO_3$, we will
also include ambient temperature, water vapor, pressure, and SZA to determine if they provide any
additional insights into $PO_3$ estimates.



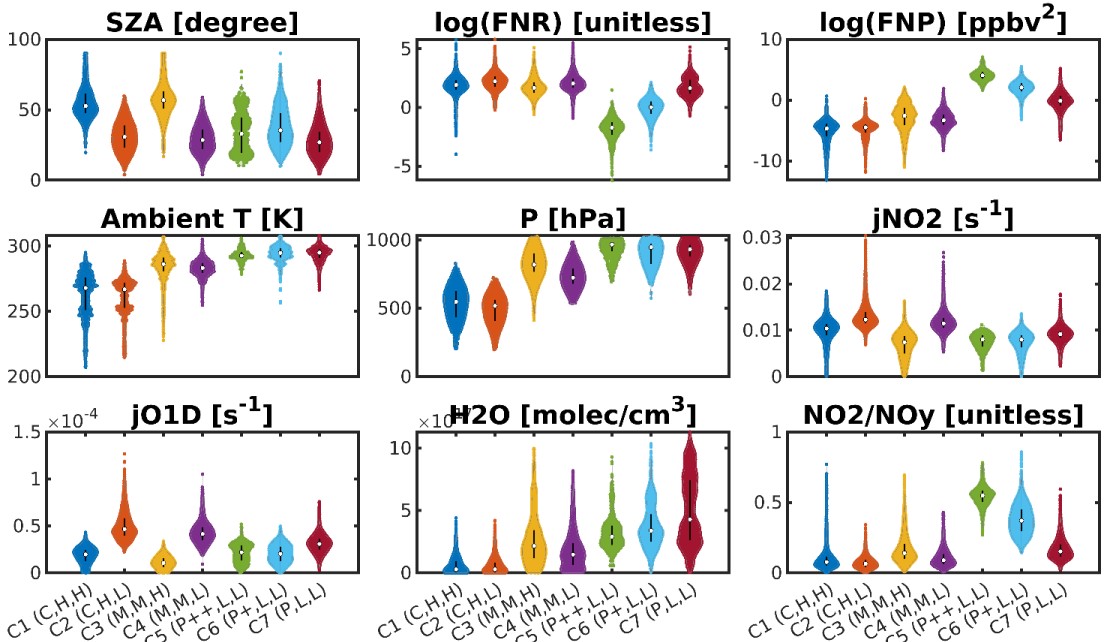


**Figure 4.** The violin plots of six different parameters coming from the box model clustered into seven distinct categories. Each cluster is described by three labels: air pollution levels (C: clean, M: moderately clean, P: moderately polluted, P+: polluted, P++: extremely polluted), altitude (H: high, M: medium, L: low), and SZA (H: high, L: low). The white dot is the median and the bars explain the 75th and 25th percentiles.

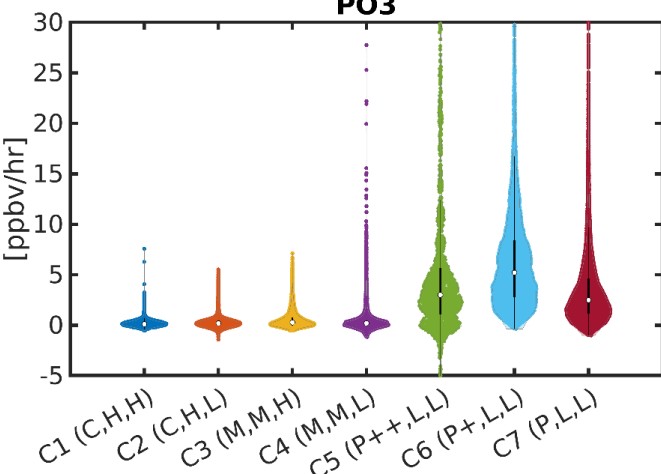

**Figure 5.** The corresponding violin plots of simulated $PO_3$ for the seven clusters described in Figure 4. The lowest $PO_3$ is seen in remote regions (C-M) where ozone precursors are minimal. The highest $PO_3$ does not happen in the most polluted region (P++) resulting from the non-linear ozone chemistry.





### 4.3. Estimates of PO₃

*4.3.1. LASSO coefficients*

Armed with a procedure that finds the important features in a linear model (Section 3.1), we now explore utilizing LASSO for $PO_3$ estimation. We are leveraging all data points generated by the observationally-constrained box model from various atmospheric composition campaigns. Among the selected variables shown in Figure 2, the LASSO algorithm assigns zero coefficients to SZA, pressure, temperature, and water vapor, indicating that they offer less valuable information compared to other variables. This decision was made by systematically adjusting the regularization factor within a 10-fold cross-validation framework to identify the optimal factor that strikes a balance between solution variance and prediction bias. As a result, the LASSO algorithm suggests that HCHO, $NO_2$, $jNO_2$, and $jO^1D$ contain sufficient information to accurately predict $PO_3$ for the most part.

Table 1 provides the intercepts and the corresponding coefficients for four different regions separated by FNR. While we do not expect for a statistical model to fully single out the "cause and effect" relationship between explanatory variables and the target, we note that it has some basic understanding of ozone chemistry; the HCHO coefficients increase as moving towards smaller FNRs (i.e., more VOC-sensitive). The same tendency is evident with respect to $NO_2$ and larger FNRs (i.e., more $NO_x$-sensitive). The negative coefficient of $NO_2$ in regions having FNR≤1.5, implies some levels of non-linear feedback embedded in this parameterization. Both $jNO_2$ and $JO^1D$ have positive coefficients throughout the chemical conditions, suggesting that more photolysis rates accelerate $PO_3$. $JO^1D$ has a smaller effect than $jNO_2$ on $PO_3$ over remote regions (FNR≥3.5) perhaps because of redundant information available compared to $jNO_2$.

**Table 1.** Calibrated coefficients derived from the LASSO estimator using seven atmospheric composition aircraft campaigns.

| Group | Criteria for FNR | Intercept | HCHO [ppbv] | NO₂ [ppbv] | jNO₂×10³ [s⁻¹] | jO¹D×10⁶ [s⁻¹] |
|---|---|---|---|---|---|---|
| 1 | FNR≤1.5 | -1.98 | 1.85 | -0.14 | 0.12 | 0.09 |
| 2 | 1.5<FNR<2.5 | -3.38 | 1.79 | 0.98 | 0.19 | 0.07 |
| 3 | 2.5<FNR<3.5 | -3.27 | 1.07 | 3.48 | 0.21 | 0.03 |
| 4 | FNR≥3.5 | -1.63 | 0.41 | 6.54 | 0.11 | 0.01 |

*4.3.2. Validation of PO₃ predictions*

The validation of $PO_3$ prediction against the box model results is performed in threefold with an increasing stringency order: i) using all data points used in the LASSO algorithm, ii) by random dropping data points, and iii) by dropping each air quality campaign from the LASSO estimation and using its data as benchmark.

Figure 6a shows the scatterplot of predicted $PO_3$ against the box model for all data points used to estimate the coefficients described in Section 4.3.1. Despite the algorithm's simplicity, we can recreate more than 88% of the variance in $PO_3$ with negligible absolute bias. This has an important indication that our scientific problem is not overly complex. There is less than 30% bias with respect to the mean absolute bias of the prediction. The positive offset and a slope smaller than one indicate a mild underestimation (overestimation) of $PO_3$ in polluted (clean) regions. Figure 6b shows the same analysis for 20,000 randomly chosen data points (~15% of the total) that we purposefully dropped from the LASSO estimation to gauge if the predictor model can replicate numbers for points not used during the training. We find almost identical statistics for these points, suggesting that the prediction stays robust for points outside the training data set. However, the most stringent method is to drop each campaign data set entirely to understand where the prediction model struggles most.





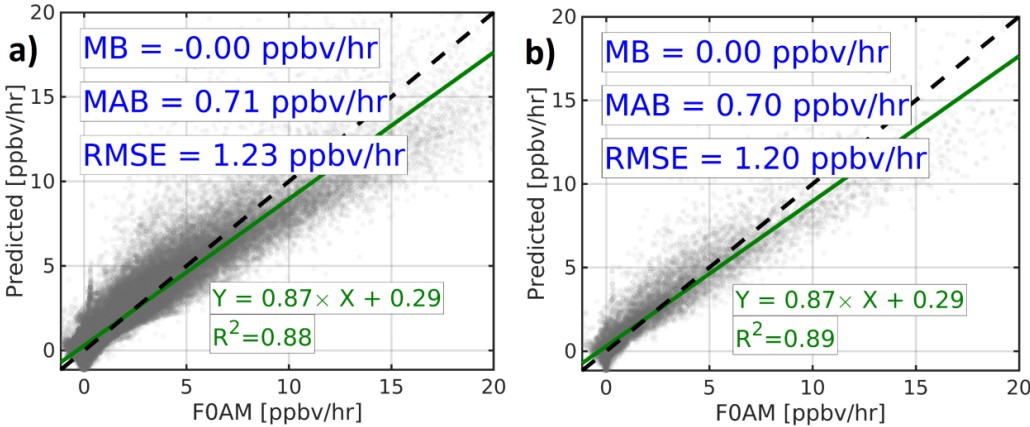


**Figure 6.** Scatterplots comparing observationally-constrained F0AM model PO$_3$ and the predictions based
on the proposed algorithm for (a) all data points and (b) 20,000 randomly-dropped data points as
benchmarks. Despite the simplicity of the algorithm, we can reproduce a large variance in PO$_3$ using only
four explanatory variables.

Figure 7 shows several subplots pertaining to dropped campaigns from the analysis. Immediately
evident is that our PO$_3$ estimation has considerable skills at capturing PO$_3$ for most polluted cases, including
DISCOVER-AQs, KORUS-AQ, and SENEX without using their individual datasets. This provides
convincing evidence about a high degree of generalizability of the predictor. However, the model has a
reduction in performance in INTEX-B for PO$_3$ <1 ppbv/h. Moreover, the model prediction power is
consistently poor for ATOMs where a significant fraction of airsheds were samples in pristine areas. We see
such poor performance for PO$_3$<1 ppbv/hr for other campaigns such as KORUS-AQ. Therefore, it is
difficult to have confidence in the predictor power of the model in remote regions, which may be caused
by the lack of inclusion of HOx, halogens, and H$_2$O in the fit, as they can become an important sink for
tropospheric ozone in those areas (Simpson et al., 2015). Nonetheless, while our predictive accuracy
remains poor for this specific subset of the data, the practical utility and significance of this specific region
(i.e., pristine areas) for air quality applications are notably limited. Given these results, we limit our
predictions to PO$_3$>1 ppbv/hr for the subsequent analyses.

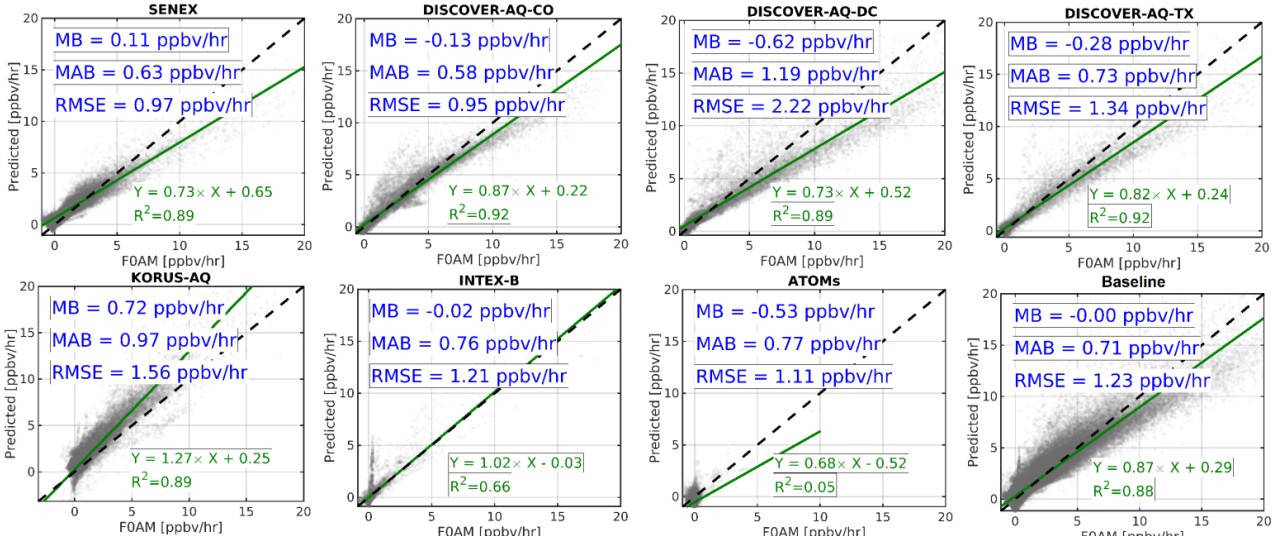

**Figure 7.** Same as Figure 6b, but each campaign is dropped from the LASSO estimation and subsequently used as an independent benchmark. The designed algorithm has shown a high degree of skill at predicting PO$_3$ in polluted regions; however, it performs poorly in pristine areas.

### 4.3.3. TROPOMI NO$_2$ and HCHO validation

To build confidence in our quantitative application of TROPOMI data for PO$_3$ estimates, we validate the daily tropospheric NO$_2$ and total HCHO columns against MAX-DOAS and FTIR observations based upon the validation framework outlined in Vigouroux et al. (2020) and Verheolst et al. (2021). Both paired datasets have been expanded to late 2023 showing a fuller picture of TROPOMI error characterization compared to former studies. Figure 8 shows the comparison of daily TROPOMI, the benchmarks and the optimal fit associated with their errors for the period of 2018-2023.

In the context of tropospheric NO$_2$ comparison, we observe a slope smaller than one (~ 0.66) with a positive offset (0.32 ×10$^{15}$ molec/cm$^2$). This tendency has been repeatedly documented in various studies for various satellites or benchmarks (e.g., Griffin et al., 2019; Choi et al., 2020; Verhoelst et al. 2021; van Geffen et al., 2022). A slope smaller than one, originating from unresolved systematic biases, implies that TROPOMI is biased-low in polluted regions. A slight positive offset suggests that TROPOMI NO$_2$ is biased-high in remote regions. The errors of slope and the offset are relatively small, evidence of the robustness of the optimal fit against the dataset variance. Nonetheless, we will incorporate them into Eqs 2 and 3 to take the adjustment error into consideration.

Despite the inherent difficulty in obtaining HCHO observations from the UV-Vis imagery (González Abad et al., 2019), the HCHO comparison exhibits a good alignment with benchmarks. Like the previous comparison, the slope is smaller than one (~0.59) and the offset is positive (~0.9 ×10$^{15}$molec/cm$^2$) agreeing within 10% with studies done by Vigouroux et al. (2020) and De Smedt et al. (2021). Consequently, we will consider the fit errors and adjust all VCDs based on the slope and the offset obtained from this comparison.





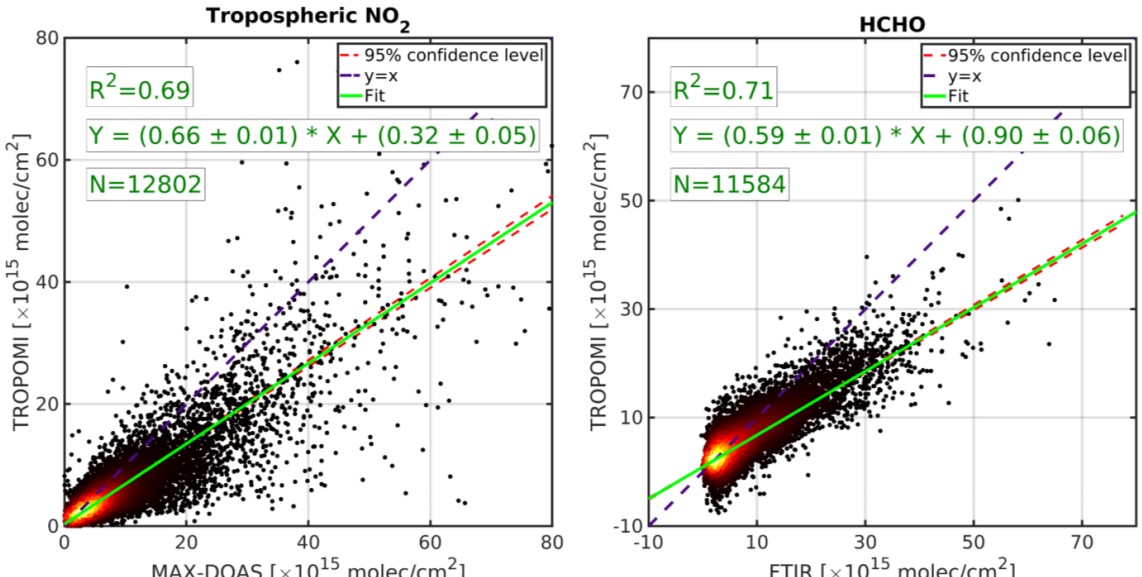

**Figure 8.** The comparison of TROPOMI tropospheric $NO_2$ and MAX-DOAS (left) and TROPOMI HCHO and FTIR (right). The data points cover the period of 2018-2023. Both errors of in-situ measurements and TROPOMI are considered in the fit. The data curation procedure has been discussed in Verhoelst et al. (2021) and Vigouroux et al. (2020). The slope smaller than one suggests that both HCHO and $NO_2$ retrievals are underestimated in polluted regions.

*4.3.4. Maps of $PO_3$ across various regions and qualitative description*

Taking advantage of the wealth of bias-corrected TROPOMI observations, we present the first-ever reported $PO_3$ maps at 0.1×0.1 degrees in the PBL in July 2019 across various geographic regions. Moreover, because of the explicit nature of our algorithm, it is straightforward to break down the contributors of $PO_3$ to gather insights into how each driver has shaped the distribution of $PO_3$. Therefore, in addition to $PO_3$ maps, we will show the magnitudes of various drivers of $PO_3$ including $NO_2$, HCHO, and FNR concentrations in the PBL region, the sum of scaled $jO^1D$ and $jNO_2$ values, along with their contributions to $PO_3$. It is worth noting that these maps are only a snapshot of $PO_3$ whose precursors can have large interannual and interdecadal variability caused by meteorology, chemistry, and emissions. A discussion on each region follows:

*Africa and the Middle East* – Figure 9 illustrates the significant rates of $PO_3$ over the region, particularly concentrated over major cities such as Tehran (Iran), Cairo (Egypt), Riyadh (Saudi), Baghdad (Iraq), Algiers (Algeria), and Johannesburg (South Africa). These urban areas consistently experience poor air quality episodes (e.g., Chaichan et al., 2016; Belhout et al., 2018; Yousefian et al., 2020; Thompson et al., 2014; Boraiy et al., 2023; Choi and Souri et al. 2015a). The biomass burning activities in Africa (see Figure 1 in Roberts et al., 2009) significantly contribute to the high rates of $PO_3$. Moreover, we see accelerated $PO_3$ over the Persian Gulf, a region housing oil and gas production facilities, leading to high $PO_3$ in the region (Lelieveld et al., 2009; Choi and Souri et al. 2015a). Figure 10 shows $NO_2$ and HCHO concentrations are highly correlated in the Middle East ($r$=0.82) due to co-emitted $NO_x$ and VOC emissions, predominantly from anthropogenic sources. Over the whole region, HCHO and $NO_2$ concentrations are only moderately correlated ($r$=0.61). This is because there is strong spatial heterogeneity associated with $NO_x$ and VOC emissions over Africa that are not spatially correlated. One possible explanation for this could be the emission dependence on the type of fire combustion in Africa (van der Velde et al., 2021) and the location



of biogenic isoprene emissions (Marais et al., 2014). For the most part, FNRs tend to fall in ranges above
>3.5 (LASSO group 4, highly $NO_x$-sensitive). However, lower FNRs are prevalent in the core of cities due
to elevated $NO_x$ emissions. The contributions of HCHO to $PO_3$ occur predominantly over areas with low
FNRs. These results suggest that $NO_x$ emissions dictate the location of maximum VOC contributions to
$PO_3$. The contribution of $NO_2$ to $PO_3$ behaves non-linearly with negative values at the core of cities such as
Johannesburg and Tehran (Figure S2). Photolysis rates are high over low SZA and bright surface albedo
(i.e., arid land). Accordingly, photolysis rates exhibit a latitudinal gradient in response to changes in SZA.
Greater contributions of photolysis rates to $PO_3$ are observed in areas with low FNRs, as determined by the
LASSO estimator (Table 1).

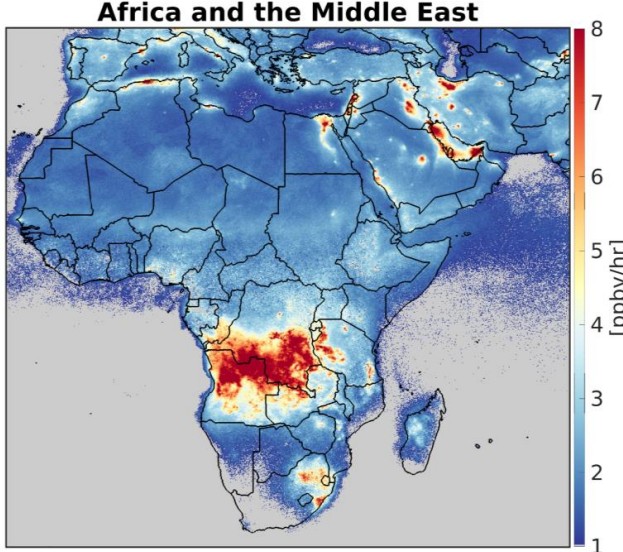


**Figure 9.** The spatial distribution of $PO_3$ within the PBL region averaged over July 2019 in Africa and the
Middle East. $PO_3$<1 ppbv/hr is masked due to the algorithm deficiencies. Accelerated $PO_3$ can be seen over
major cities and biomass burning activities in Africa.

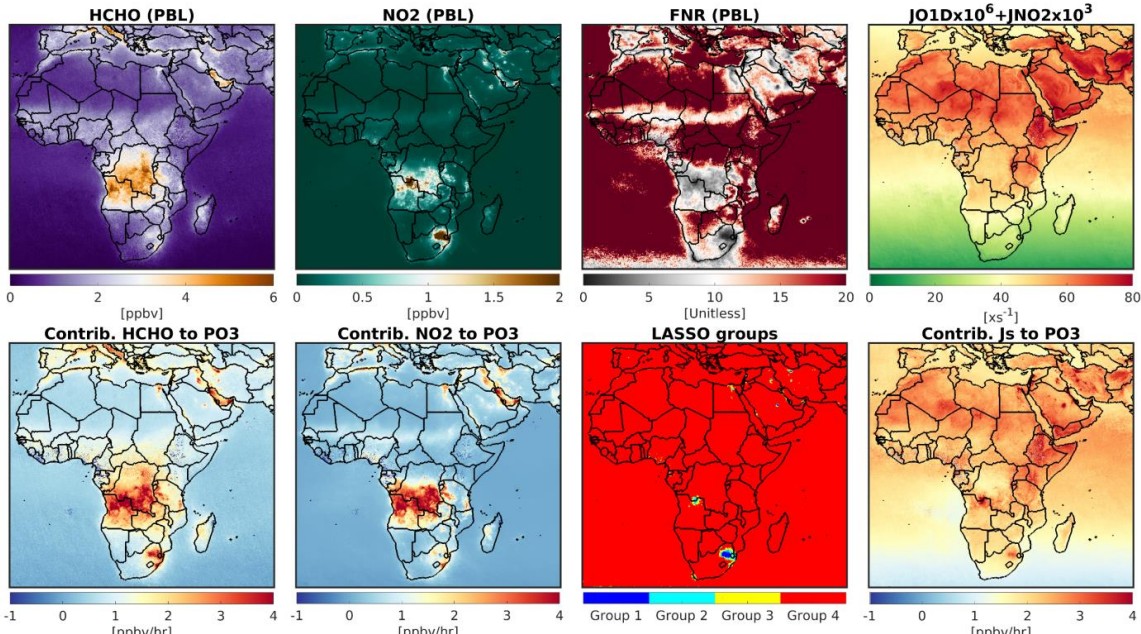


**Figure 10.** (first row) PBL concentrations of HCHO, NO$_2$, FNR and sum of scaled jO$^1$D and jNO$_2$ derived from TROPOMI and models in July 2019; (second row) the contributions of HCHO, NO$_2$, and photolysis rates to PO$_3$, along with the defined LASSO ozone production sensitivity regimes for PO$_3$ estimates.

*Contiguous United States* – New York City, Los Angeles (LA), the San Francisco Bay area, and Lake Michigan areas all experience accelerated PO$_3$ in July 2019, as shown in Figure 11. All the regions fall into non-attainment regions (marginal to extreme) with respect to ozone standards and have been immensely studied (Wu et al., 2024; Kim et al., 2022; Stainer et al., 2021). While it requires several physical processes, such as vertical and horizontal transport, to translate these PO$_3$ rates into ozone concentrations, applying this product in locating the hotspot of ozone polluters shows promise. Except for LA, the vast majority of CONUS fall into large FNRs (>3.5), making NO$_2$ levels largely shape the spatial distribution of PO$_3$ (Figure 12). HCHO levels are found to be relatively large over LA, causing PO$_3$ to increase due to its greater sensitivity to VOCs. In addition to high levels of HCHO and NO$_2$ in several Californian regions, accelerated photochemistry caused by the bright surface albedo enhances PO$_3$.



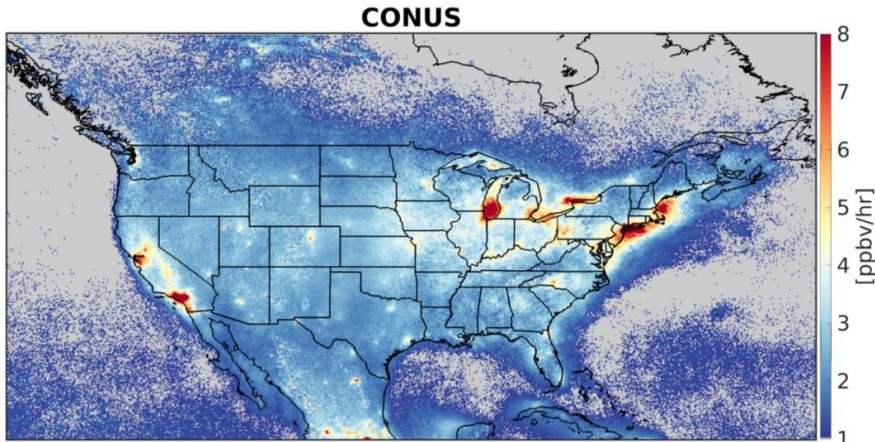

**Figure 11.** Same as Figure 9 but for CONUS. Elevated PO$_3$ prevails over various areas such as New York City, Los Angeles, San Francisco Bay area, and Lake Michigan.

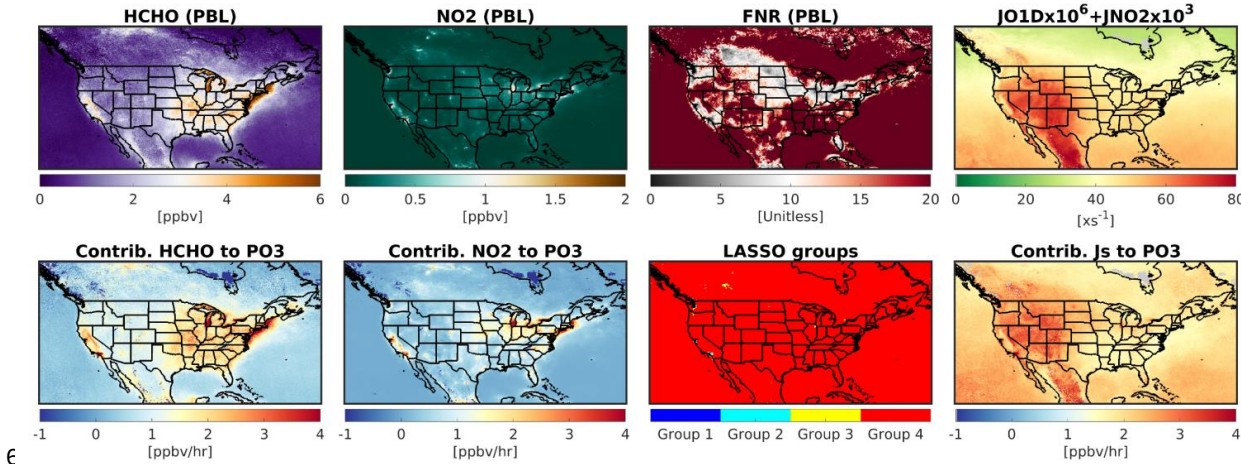

**Figure 12.** Same as Figure 10 but for CONUS.

*East and Southeast Asia* – Figure 13 shows extremely accelerated PO$_3$ values over East Asia, particularly over North China Plain, Yangtze River Delta, Pearl River Delta, and Seoul. These regions have experienced severely degraded air quality with respect to ozone (Souri et al., 2020a,b; Li et al., 2019; Colombi et al., 2023; Schroeder et al., 2020; Wang et al., 2017; Zhang et al., 2007). In southeast Asia, Hanoi (Vietnam), Kuala Lumpur (Malaysia), and Jakarta (Indonesia), undergoing heightened PO$_3$ as well, have received less attention in literature (Ahamad et al., 2020; Kusumaningtyas et al., 2024; Sakamoto et al., 2018). Figure 14 suggests that the chemical condition of many regions in China and South Korea, falling within the transitional regimes (LASSO group 2 and 3, 1.5<FNR<3.5), has made them susceptible to high PO$_3$ levels due to concurrent high concentrations of HCHO and NO$_2$. Moreover, photochemistry appears to be active throughout the region.



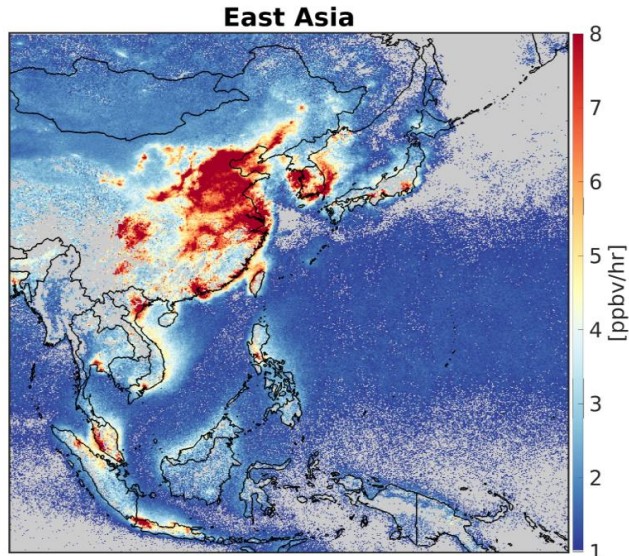

**Figure 13.** Same as Figure 9 but for east and southeast Asia. Because of heightened amount of photochemistry, $NO_2$, and HCHO, we observe accelerated $PO_3$ throughout the majority of the cities in East and Southeast Asia.

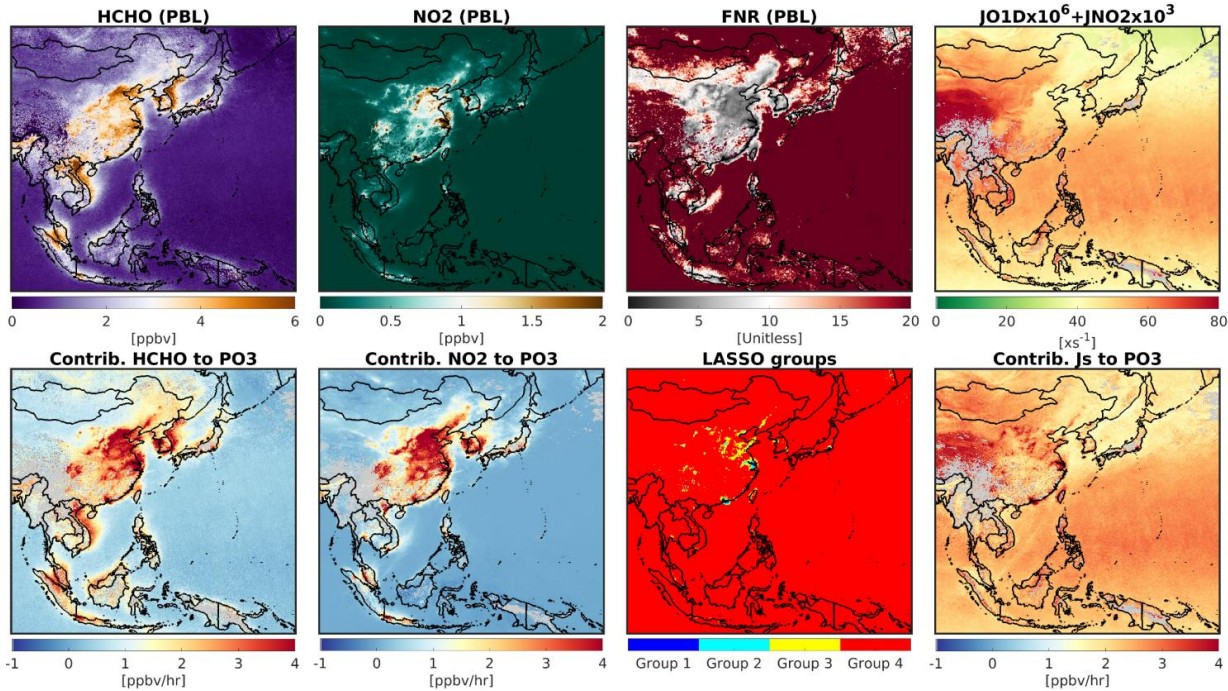



**Figure 14.** Same as Figure 10 but for east and southeast Asia.
*Europe* – Figure 15 reveals high $PO_3$ over Benelux, Po Valley (Italy), and several major cities such as
Barcelona (Spain) and Rome (Italy). Benelux has the largest hotspot of $PO_3$ in the region (e.g., Zara et al.,
2021). A significant portion of England, Benelux, fall into VOC-sensitive, or the transitional regime
(FNR<2.5) shown in Figure 16. Because of diminished photochemistry in these high latitude regions, we
do not see significant PBL concentrations of HCHO in order for $PO_3$ to be as high as the previous areas;
moreover, the non-linear $NO_x$ feedback has led to negative contributions of $NO_2$ to $PO_3$ in several cities
such as London. In general, low photolysis rates compared to the previous regions have made most of
Europe less prone to elevated $PO_3$.

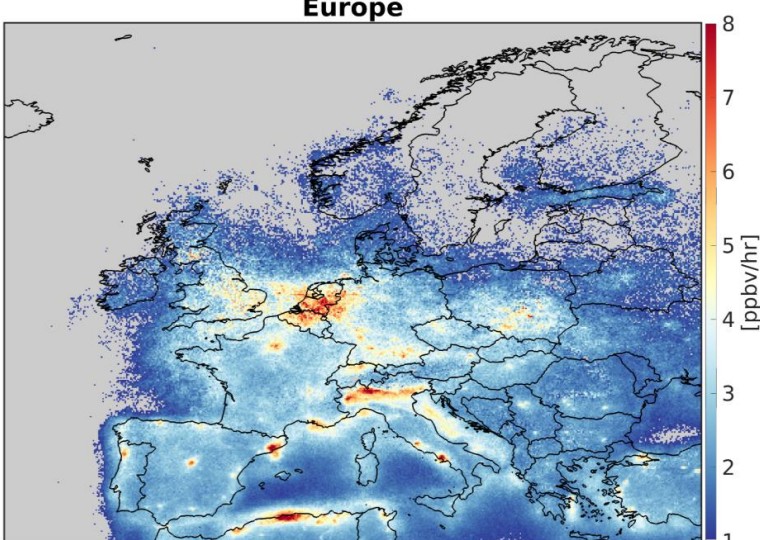


**Figure 15.** Same as Figure 9 but for Europe. Because of reduced photochemistry, $PO_3$ values tend to be
smaller than the previous cases. Benelux has experienced the highest $PO_3$ in this region.





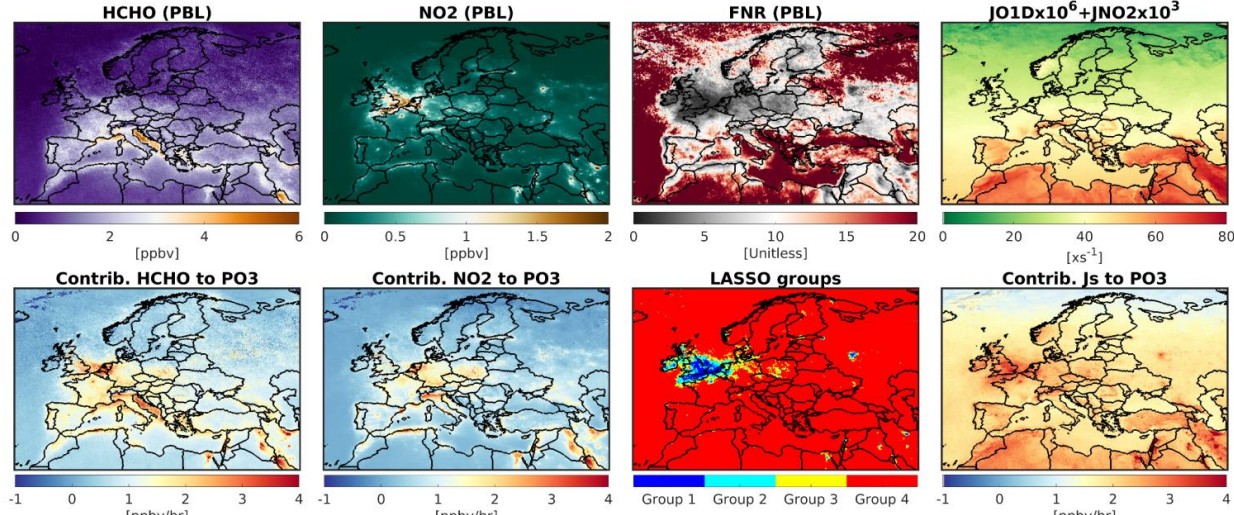

**Figure 16.** Same as Figure 10 but for Europe.

*4.3.5. Seasonality of PO₃ over the Middle East*

It is attractive to study the seasonal variations in the contributors to PO₃ over several major cities because the PO₃ drivers' seasonality can vary from location to location. We decide to focus on several Middle Eastern countries that have experienced rapid growth and degraded air quality: Cairo (Egypt), Ghaza (Palestine), Baghdad (Iraq), Riyadh (Saudi Arabia), Tehran (Iran), and the Persian Gulf region. We illustrate the seasonality of four major contributors to PO₃ including $NO_2$, HCHO, $jNO_2$, and $jO^1D$ in 2019 in Figure 17.

The levels of HCHO (a proxy for VOCs) consistently have the greatest impact on PO₃ throughout the year in these regions. Specifically, both Baghdad and Tehran experience high levels of HCHO even during colder months, which can be observed using TROPOMI. This suggests that regulations targeting the reduction of man-made VOC emissions should be prioritized in this region. PO₃ levels over Cairo, Gaza, Baghdad, and the Persian Gulf peak during summertime, while Tehran experiences a comparable peak in the autumn due to increased VOC emissions. Additionally, we notice a decrease in PO₃ levels over the Persian Gulf and Riyadh in July, possibly due to a decline in HCHO contributions caused by meteorological factors. Even though $NO_2$ concentrations decline in summertime due to shorter lifetime against OH, the higher amount of HCHO makes PO₃ more sensitive to $NO_2$ in this season. Ghaza shows the least seasonal variation among these regions, likely due to consistently active photochemistry throughout the year.



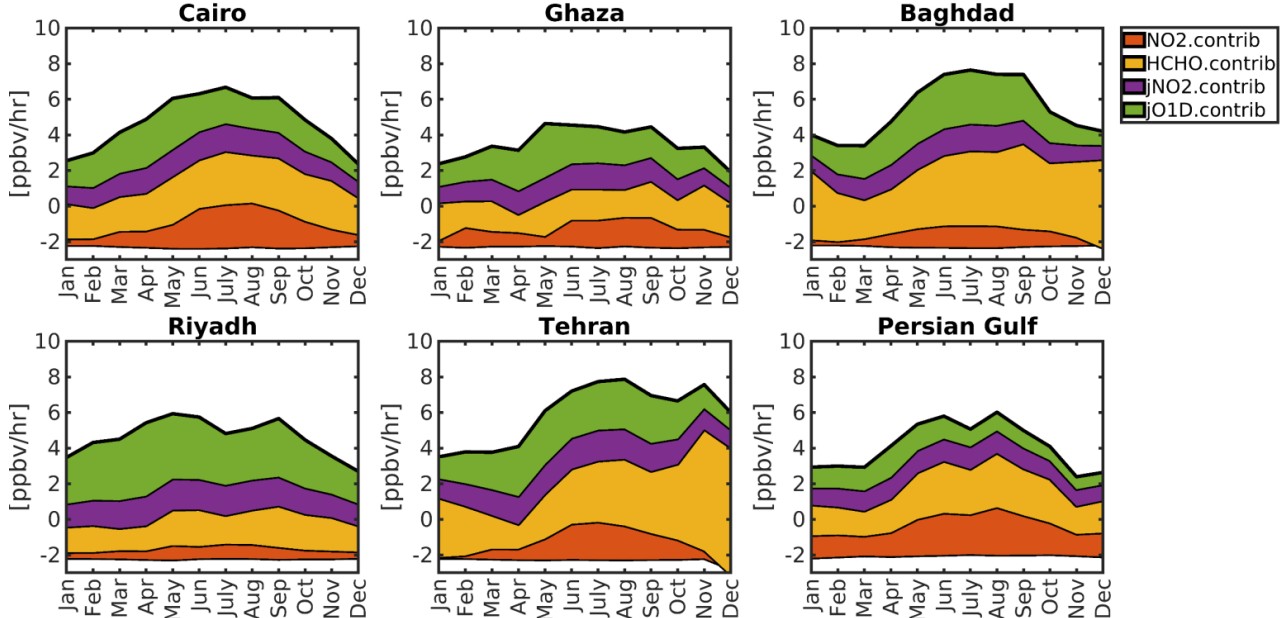

**Figure 17.** The contributions of $NO_2$, HCHO, $jNO_2$, and $jO^1D$ to the PBL $PO_3$ for several major regions in the Middle East. The data is based on 2019 TROPOMI observations. $PO_3$ tends to spike around the summer due to increased HCHO, higher sensitivity of $PO_3$ to $NO_x$, and enhanced photochemistry. However, Tehran shows a second peak in autumn due to unusual high values of HCHO.

### 4.3.6. The effect of satellite errors on PO₃

Satellite retrieval errors have been identified as the primary obstacle to achieving a robust understanding of ozone chemistry using HCHO and $NO_2$ data (Souri et al., 2023; Johnson et al., 2023); therefore, generating uncertainty maps is crucial for informing the scientific community about the credibility of our $PO_3$ estimates. In this study, we utilize the equations outlined in Section 2.2.1 to propagate the errors of HCHO and $NO_2$ retrievals to the final $PO_3$ estimates. We achieve this by recalculating the $PO_3$ value for a given pixel 10,000 times, with each recalculation based on a sample drawn from a normal distribution with a standard deviation equal to the satellite total error. The standard deviation of these samples offers a good approximation of the impact of satellite errors on $PO_3$ estimates.

Figure 18 illustrates the maps of $PO_3$ absolute and relative errors over the targeted regions in the course of the month of July. The errors of $PO_3$ estimates tend to be markedly high (100-300%) in remote regions where the trace gas signals are small. However, the $PO_3$ errors are within 40-60% in polluted regions where the signals are larger. Currently, the absence of absolute measurements of $PO_3$ at this vast spatial coverage makes it challenging to judge the severity of these errors for $PO_3$ applications. Nonetheless, any application based on this product should be recalculated within the reported errors through a Monte-Carlo to gauge the significance of the outcome.

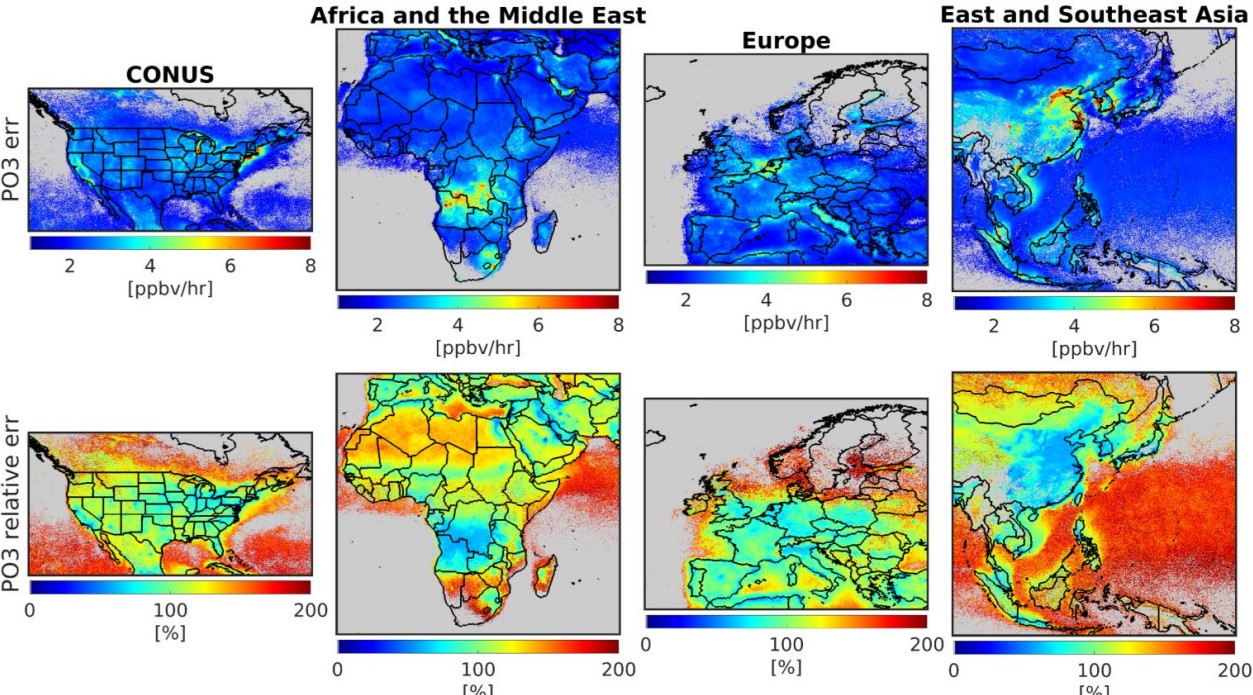

**Figure 18.** The influence of the satellite errors on PO$_3$ estimates (absolute and relative) over four major regions tackled in this work. The errors are based on monthly-averaged TROPOMI errors. The errors tend to be mild over polluted regions (40-60%) but they can exceed above 100% over pristine ones.

## 5. Conclusion

Providing data-driven and integrated maps of ozone production rates (PO$_3$) using remote sensing sensors enabled us to generate the first satellite-derived product of this kind, offering extensive spatial coverage with significant applications in atmospheric chemistry. This data has indeed extended the use of formaldehyde (HCHO) over nitrogen dioxide (NO$_2$) ratios (FNR) beyond their current role. Through this product, we can shed light on the effects of emission regulations, wildfires, widespread lockdown, wars, and economic recessions on PO$_3$ levels. Furthermore, given the long-term records of satellite observations (e.g., OMI since 2005 and TROPOMI since 2018), this product can inform emission regulators about locally-produced ozone hotspots, and ultimately, enhance our understanding of the spatiotemporal variability of ozone formation for over two decades.

In this study, we generated maps of within the planetary boundary layer (PBL), constrained by bias-corrected TROPOspheric Monitoring Instrument (TROPOMI) observations, using a piecewise regularized regression model. This model was calibrated using a blend of data from a comprehensive suite of aircraft observations and a well-characterized box model. These maps, produced for various regions, allowed us to identify hotspots of locally-produced ozone pollution with unprecedented resolution. Our findings indicated that numerous urban areas in the Middle East, East Asia, and Southeast Asia exhibit accelerated PO$_3$ rates (>10 ppbv/hr), attributed to high levels of anthropogenic nitrogen oxides (NO$_x$ = NO + NO$_2$), volatile organic compounds (VOCs), and active photochemistry. In contrast, such elevated PO$_3$ levels were less prevalent in the United States and Europe, with exceptions including Los Angeles, New York City, and the



entire region of the Benelux. Additionally, biomass burning activities in Africa contributed to significant $PO_3$ levels across extensive areas. Seasonality of $PO_3$ peaked around the summer for several regions in the Middle East because of active photochemistry and concurrent large HCHO and $NO_2$ levels; however, Tehran experienced elevated $PO_3$ in the autumn due to large HCHO values possibly produced from anthropogenic emissions.

The production of these maps relied heavily on a robust training dataset. To this end, we incorporated an extensive array of aircraft observations from multiple atmospheric composition campaigns, including DISCOVER-AQ, KORUS-AQ, INTEX-B, ATOM, and SENEX, into the Framework for 0-D Atmospheric Modeling (F0AM) photochemical box model. The box model demonstrated a high level of correspondence ($R^2 > 0.6$, with minimal biases) between several unconstrained compounds (e.g., HCHO, OH, $HO_2$, PAN, NO, and $NO_2$) and their observed counterparts, indicating its effectiveness in understanding local ozone chemistry. Utilizing a classification algorithm applied to the data obtained from the constrained box model, we identified HCHO, $NO_2$, their ratio (known as FNR), photolysis rates, and, to some extent, meteorological factors as good candidates for reproducing $PO_3$ variability and magnitudes.

Subsequently, we employed a piecewise linear model known as LASSO, which is capable of feature selection by eliminating unimportant inputs, to parameterize $PO_3$. A key component of this parameterization was the use of FNR to empirically linearize the non-linear ozone chemistry. The LASSO algorithm indicated that more than 88% of the variance in $PO_3$ could be reproduced with low bias using only five parameters: FNR, HCHO, $NO_2$, $jNO_2$ (photolysis rates for $NO_2$ + hv), and $jO^1D$ (photolysis rates for $O_3$ + hv). This parameterization demonstrated remarkable performance for the majority of air parcels collected in moderately to extremely polluted regions ($PO_3 > 1$ ppbv/hr). However, it performed poorly in pristine regions due to the exclusion of certain significant ozone loss pathways, such as $HO_x$ (OH+$HO_2$), which are more challenging to predict.

Fortunately, TROPOMI provided critical data to enhance the representation of FNR, HCHO, $NO_2$, $jNO_2$, and $jO^1D$. We utilized TROPOMI's viewing geometry, UV surface albedo, and total ozone overhead from a model to predict $jNO_2$ and $jO^1D$ using look-up tables derived from NCAR's TUV model. To convert TROPOMI tropospheric $NO_2$ and HCHO columns to their PBL mixing ratios, we employed the MERRA2GMI global transport model, extensively used in various studies. However, the coarse resolution of this model might have introduced underrepresentation issues, which could be mitigated by using higher spatial resolution models in future research.

To address the biases associated with TROPOMI observations, we updated comparisons from Verhoelst et al. (2021) and Vigouroux et al. (2020) with a larger dataset of paired TROPOMI and FTIR/MAX-DOAS measurements. TROPOMI retrievals significantly underestimated HCHO and $NO_2$ magnitudes in polluted regions (slope ~0.6 - 0.7) and moderately overestimated them in pristine areas. These biases were corrected using regression lines, enabling a relatively unbiased application of the data.

To build confidence in our product, we propagated TROPOMI HCHO and $NO_2$ errors to $PO_3$ estimates using a Monte Carlo approach. Results indicated that $PO_3$ estimates were highly uncertain (100-300%) in clean regions due to a low trace gas signal in TROPOMI retrievals. However, in polluted regions, the errors were more moderate (40-60%) due to the stronger signal.

Over the years, extensive efforts have been devoted to measuring various critical atmospheric compounds globally, developing robust atmospheric models, and enhancing satellite retrievals along with their benchmarks. These advancements have enabled us to estimate $PO_3$ maps within the planetary boundary layer (PBL) using comprehensive data. Nonetheless, it is crucial to acknowledge some limitations of our work, many of which are the focus of ongoing research within our team:



i) The direct measurement of PO$_3$ using specialized instruments (Cazorla and Brune, 2010;
Sadanaga et al., 2017; Sklaveniti et al., 2018) is lacking in most atmospheric composition datasets, limiting
our ability to fully understand the effects of assumptions (such as the exclusion of heterogeneous chemistry)
made in the box model on PO$_3$.
ii) There is potential for improvement in the parameterization process by employing more
sophisticated algorithms, such as neural networks, which could increase the variance explained in the
predicted PO$_3$.
iii) The conversion of satellite column data to PBL mixing ratios requires error characterization
and the use of finer-resolution models that are comparable in size to the PO$_3$ grid boxes.
iv) Partially cloudy pixels and aerosols can affect photolysis rates, which should be considered in
future parameterization efforts.
Despite these limitations, our novel product offers an asset to the atmospheric science community.
It provides a more comprehensive understanding of PO$_3$, sheds light on the complexities associated with
spatiotemporal variability associated with the non-linear ozone chemistry at a large domain, enhances
confidence in high-resolution maps of locally-produced ozone hotspots, and facilitates the investigation of
disparities and inequalities in regions where environmental justice is a concern.

## Financial Support

This study is funded by NASA's ACMAP project (grant no. 80NSSC23K1250). The measurements at
Paramaribo have been supported by the BMBF (German Ministry of Education and Research) in project
ROMIC-II's subproject TroStra (01LG1904A). The NDACC FTIR stations Bremen, Garmisch, Izaña, Ny-
Ålesund, Paramaribo, and Karlsruhe have been supported by the German Bundesministerium für Wirtschaft
und Energie (BMWi) via DLR5 under grants 50EE1711A, B, and D. The measurements and data analysis
at Bremen are supported by the Senate of Bremen. The NCAR FTS observation programs at Thule, GR,
Boulder, CO, and Mauna Loa, HI, are supported under contract by the National Aeronautics and Space
Administration (NASA). The National Center for Atmospheric Research is sponsored by the National
Science Foundation. The Thule effort is also supported by the NSF Office of Polar Programs (OPP).
Operations at the Rikubetsu and Tsukuba FTIR sites are supported in part by the GOSAT series project. The
Paris TCCON site has received funding from Sorbonne Université, the French research center CNRS, and
the French space agency CNES. The Jungfraujoch FTIR data are primarily available thanks to the support
provided by the F.R.S. FNRS (Brussels), the GAW-CH program of MeteoSwiss (Zürich), and the HFSJG.ch
Foundation (Bern). IUP-Bremen ground-based measurements are funded by DLR-Bonn and received
through project 50EE1709A. KNMI ground-based measurements in De Bilt and Cabauw are partly
supported by the Ruisdael Observatory project, Dutch Research Council (NWO) contract 184.034.015, by
the Netherlands Space Office (NSO) for Sentinel-5p/TROPOMI validation, and by ESA via the EU CAMS
project.

## Competing interests

Bryan N. Duncan is a member of the editorial board of Atmospheric Chemistry and Physics

## Acknowledgements

We thank all principal investigators, pilots, and managers who collected the aircraft data used in our
research and made them publicly available. We thank the FTIR HCHO measurement team of Thomas
Blumenstock, Martine De Mazière, Michel Grutter, James W. Hannigan, Nicholas Jones, Rigel Kivi, Erik
Lutsch, Emmanuel Mahieu, Maria Makarova, Isamu Morino, Isao Murata, Tomoo Nagahama, Justus



Notholt, Ivan Ortega, Mathias Palm, Amelie Röhling, Matthias Schneider, Dan Smale, Wolfgang Stremme,
Kim String, Youwen Sun, Ralf Sussmann, Yao Té, and Pucai Wang. We thank the Meteorological Service
Suriname and Cornelis Becker for their support. The MAX-DOAS data used in this publication were
obtained from Alkis Bais, John Burrows, Ka Lok Chan, Michel Grutter, Cheng Liu, Hitoshi Irie, Vinod
Kumar, Yugo Kanaya, Ankie Piters, Claudia Rivera-Cárdenas, Andreas Richter, Michel Van Roozendael,
Robert Ryan, Vinayak Sinha, and Thomas Wagner. Fast delivery of MAX-DOAS data tailored to the S5P
validation was organized through S5PVT AO project NID-FORVAL. We thank the IISER Mohali
atmospheric chemistry facility for supporting the MAX-DOAS measurements at Mohali, India. We thank
Julie M. Nicely for providing merged ATOMs observations.
**Authors' contributions**
AHS designed and implemented the research idea, analyzed the data, made all figures, and wrote the
manuscript. TV, CV, GP, SC, and BL provided the paired TROPOMI and benchmark data. Other authors
helped with the analysis, the model setup, and interpretation.

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
