# Peer review of "Feasibility of robust estimates of ozone production rates"

_EGUsphere, 2024_

## Community Comment (CC1)

Comments by Owen R. Cooper (TOAR Scientific Coordinator of the Community Special Issue) on:

**Feasibility of robust estimates of ozone production rates using satellite observations**

Amir H. Souri, Gonzalo González Abad, Glenn M. Wolfe, Tijl Verhoelst, Corinne Vigouroux, Gaia Pinardi, Steven Compernolle, Bavo Langerock, Bryan N. Duncan, and Matthew S. Johnson

EGUsphere [preprint], https://doi.org/10.5194/egusphere-2024-1947

Discussion started: 26 July 2024; Discussion closes 29 Sept., 2024

This review is by Owen Cooper, TOAR Scientific Coordinator of the TOAR-II Community Special Issue. I, or a member of the TOAR-II Steering Committee, will post comments on all papers submitted to the TOAR-II Community Special Issue, which is an inter-journal special issue accommodating submissions to six Copernicus journals: ACP (lead journal), AMT, GMD, ESSD, ASCMO and BG. The primary purpose of these reviews is to identify any discrepancies across the TOAR-II submissions, and to allow the author teams time to address the discrepancies. Additional comments may be included with the reviews. While O. Cooper and members of the TOAR-II Steering Committee may post open comments on papers submitted to the TOAR-II Community Special Issue, they are not involved with the decision to accept or reject a paper for publication, which is entirely handled by the journal's editorial team.

**General Comments:**

TOAR-II has produced two guidance documents to help authors develop their manuscripts so that results can be consistently compared across the wide range of studies that will be written for the TOAR-II Community Special Issue. Both guidance documents can be found on the TOAR-II webpage:
https://igacproject.org/activities/TOAR/TOAR-II

*The TOAR-II Community Special Issue Guidelines*: In the spirit of collaboration and to allow TOAR-II findings to be directly comparable across publications, the TOAR-II Steering Committee has issued this set of guidelines regarding style, units, plotting scales, regional and tropospheric column comparisons, tropopause definitions and best statistical practices.

*Guidance note on best statistical practices for TOAR analyses*: The aim of this guidance note is to provide recommendations on best statistical practices and to ensure consistent communication of statistical analysis and associated uncertainty across TOAR publications. The scope includes approaches for reporting trends, a discussion of strengths and weaknesses of commonly used techniques, and calibrated language for the communication of uncertainty. Table 3 of the TOAR-II statistical guidelines provides calibrated language for describing trends and uncertainty, similar to the approach of IPCC, which allows trends to be discussed without having to use the problematic expression, "statistically significant".

**Specific Comments:**

The goal of this analysis is to develop globally consistent maps of ozone production (these maps do not depict ozone loss, or net ozone production). The authors suggest on line 595 that the ozone production maps can be used to identify regions with high levels of ozone pollution, which seems perfectly reasonable. For example, they show that the regions of New York City, Los Angeles, San Francisco Bay area, and Lake Michigan have high levels of ozone production, and these same regions are well-known for persistent ozone pollution in the summer months. However, there are many other regions across the

USA and Canada that have high ozone pollution levels, which don't seem to stand out on the map for July 2019. Perhaps this is due to just one month being shown, and perhaps other regions would stand out during other months, but without any evaluation, we don't know if this is the case. I think there is a good opportunity here to apply the ozone observations in the TOAR database to these ozone production maps to see if they do indeed capture the urban areas with the highest ozone pollution. For example, Figure S1b in the supplement to Fleming et al. (2018) (pasted below) shows the number of days per year that maximum daily 8-hour average ozone (MDA8) exceeds 70 ppb, across North America, Europe and East Asia, based on observed ozone from 2010 to 2014 (these data are from the TOAR database). A similar map could be made for July 2019 (or other months) to see if the observed ozone pollution hotspots coincide with the ozone production hot spots. The American Lung Association publishes an annual report (State of the Air) listing the urban areas in the USA with the highest ozone pollution. The most recent analysis, based on EPA ozone data for 2020-2022, lists the following urban areas with the highest ozone pollution (number of days per year when MDA8 ozone exceeds 70 ppb) (https://www.lung.org/research/sota/city-rankings/most-polluted-cities):

1. Los Angeles
2. Visalia (Central Valley)
3. Bakersfield (Central Valley)
4. Fresno (Central Valley)
5. Phoenix, AZ
6. Denver, CO
7. Sacramento (Central Valley)
8. San Diego
9. Salt Lake City, UT
10. Houston
11. Las Vegas
12. San Jose-San Francisco-Oakland
13. Dallas
14. NYC
15. El Paso, TX
16. Fort Collins, CO
17. Chicago
18. El Centro, CA
19. Reno, NV
20. Colorado Springs, CO

The 4 high ozone production regions in the USA identified by the authors are in this list of the top 20 polluted cities, but why don't the other 16 cities stand out on the ozone production map? Similarly, the Po Valley of northern Italy is the ozone hot spot for Europe, but the ozone production map gives the impression that Benelux would have higher ozone pollution levels.

**Minor Comments:**

In the first paragraph the authors make some general statements about the importance of ozone for health, vegetation and climate, but provide no references. This would be a good opportunity to cite the findings from the first phase of TOAR in three key publications, TOAR-Health (Fleming et al., 2018), TOAR-Vegetation (Mills et al., 2018) and TOAR-Climate (Gaudel et al., 2018).

Line 199
SZA is first mentioned here, but it needs to be defined

Line 679
"This data has" should be "These data have"

**References:**

Fleming, Z. L., R. M. Doherty, et al. (2018), Tropospheric Ozone Assessment Report: Present-day ozone distribution and trends relevant to human health, *Elem Sci Anth*, *6(1):12,* DOI: https://doi.org/10.1525/elementa.273

Gaudel, A., et al. (2018), Tropospheric Ozone Assessment Report:  Present-day distribution and trends of tropospheric ozone relevant to climate and global atmospheric chemistry model evaluation, *Elem. Sci. Anth., 6(1):39*, DOI: https://doi.org/10.1525/elementa.291

Mills, G., et al. (2018), Tropospheric Ozone Assessment Report: Present-day tropospheric ozone distribution and trends relevant to vegetation, *Elem. Sci. Anth*., *6(1):47*, DOI: https://doi.org/10.1525/elementa.302

[Figure]

**Figure S1b** from Fleming et al., 2018

---

## Author Comment (AC1)

**We thank Dr. Cooper for his insights and constructive comments. Our response is as follows:**

Specific Comments:

The goal of this analysis is to develop globally consistent maps of ozone production (these maps do not depict ozone loss, or net ozone production). The authors suggest on line 595 that the ozone production maps can be used to identify regions with high levels of ozone pollution, which seems perfectly reasonable. For example, they show that the regions of New York City, Los Angeles, San Francisco Bay area, and Lake Michigan have high levels of ozone production, and these same regions are well-known for persistent ozone pollution in the summer months. However, there are many other regions across the USA and Canada that have high ozone pollution levels, which don't seem to stand out on the map for July 2019. Perhaps this is due to just one month being shown, and perhaps other regions would stand out during other months, but without any evaluation, we don't know if this is the case. I think there is a good opportunity here to apply the ozone observations in the TOAR database to these ozone production maps to see if they do indeed capture the urban areas with the highest ozone pollution. For example, Figure S1b in the supplement to Fleming et al. (2018) (pasted below) shows the number of days per year that maximum daily 8-hour average ozone (MDA8) exceeds 70 ppb, across North America, Europe and East Asia, based on observed ozone from 2010 to 2014 (these data are from the TOAR database). A similar map could be made for July 2019 (or other months) to see if the observed ozone pollution hotspots coincide with the ozone production hot spots. The American Lung Association publishes an annual report (State of the Air) listing the urban areas in the USA with the highest ozone pollution. The most recent analysis, based on EPA ozone data for 2020-2022, lists the following urban areas with the highest ozone pollution (number of days per year when MDA8 ozone exceeds 70 ppb) (https://www.lung.org/research/sota/city-rankings/most-polluted-cities):

1. Los Angeles

2. Visalia (Central Valley)

3. Bakersfield (Central Valley)

4. Fresno (Central Valley)

5. Phoenix, AZ

6. Denver, CO

7. Sacramento (Central Valley)

8. San Diego

9. Salt Lake City, UT

10. Houston

11. Las Vegas

12. San Jose-San Francisco-Oakland

13. Dallas

14. NYC

15. El Paso, TX

16. Fort Collins, CO

17. Chicago

18. El Centro, CA

19. Reno, NV

20. Colorado Springs, CO

The 4 high ozone production regions in the USA identified by the authors are in this list of the top 20 polluted cities, but why don't the other 16 cities stand out on the ozone production map? Similarly, the Po Valley of northern Italy is the ozone hot spot for Europe, but the ozone production map gives the impression that Benelux would have higher ozone pollution levels.

**Response**

**Ozone production rates are only one element among several physiochemical processes determining ozone concentration. All these elements can be categorized into:**

*Chemistry (ozone production rates) + vertical transport (advection + diffusion) + horizontal transport (advection + diffusion) + cloud chemistry + dry deposition + background values*

**Of particular significance is the long lifetime of ozone ($O_X\sim$ 73 days), which, when combined with the increasing background ozone concentration by altitude, wields a significant influence on the transport effect. This influence is so pronounced that places like Denver, with their higher altitudes and thus more contribution of ozone through vertical diffusion and background ozone, do not need a significant amount of local ozone production rates to push surface ozone to an unhealthy level.**

**Some of these physiochemical processes can have conflicting signs requiring us to perform a full-chemistry modeling experiment. For instance, an expanded PBLH (thus more turbulence and less aerodynamic resistance) increases the contributions of high ozone aloft to the surface but simultaneously increases the dry deposition velocities over vegetated areas. Understanding how these conflicting contributors can cancel each other needs a model.**

**We observed a prime example of a decoupled relationship between ozone concentration and $PO_3$ in Souri et al. 2020 (https://www.sciencedirect.com/science/article/pii/S1352231020300820) over Seoul during a degraded air quality episode (June 9th, 2016). Figure 4 in that paper shows a high**

concentration of HCHO and NO₂ over Olympic Park, leading to large PO₃ (also observed in Figure 8 in [https://acp.copernicus.org/articles/19/5051/2019/acp-19-5051-2019.pdf](https://acp.copernicus.org/articles/19/5051/2019/acp-19-5051-2019.pdf)). Despite accelerated PO₃, ozone concentration was substantially low compared to suburbs. This could result from various reasons, including larger dry deposition velocities over the park and lower aircraft altitude undergoing reduced background ozone + less ozone contribution through vertical diffusion.

Our research is backed by one of the most comprehensive and well-constrained box models in our deposit, giving us a strong foundation for our claim regarding the decoupled relationship between PO₃ and ozone levels. The following figures, which contrast PO₃ (x-axis) and ozone concentration (y-axis) during the KORUS-AQ campaign, including all altitudes and limited to <1500 m, provide clear evidence of a poor correlation (r<0.1) between them:

[Figure]

These figures suggest that a linear relationship between ozone concentration and PO₃ cannot be established; therefore, we should not expect high local PO₃ rates to strictly provide authoritative explanation of non-attainment regions.

**While the decoupled relationship between ozone levels and PO₃ may appear to be a weakness for our product, we believe that it is a major strength. Due to convoluted physiochemical processes determining ozone concentration, one cannot easily attribute a trend in surface level to a specific contributor. It is because of this reason that we have to make several assumptions to rule out specific contributors (e.g., limiting observations to nighttime mountainous region) or to use various model experiments under various realizations. The advantage of our product lies in the fact that it provides a robust piece of information about the chemistry component largely influenced by local emissions. If one observes elevated surface ozone while moderate/low PO₃, our product signals the need for further investigation into other components.**

**We will eventually release the data for the period of 2005-2024 using OMI and TROPOMI to have a full picture of hotspots of PO₃. We do not wish to suggest that our product is sufficient to explain surface ozone variability. Therefore, we limit our response to this feedback by providing more caveats so as not to oversell the product.**

**Modifications**

**In Section 4.3.4. right after mentioning non-attainment region, we removed this part: "While it requires several physical processes, such as vertical and horizontal transport, to translate these PO₃ rates into ozone concentrations, applying this product in locating the hotspot of ozone polluters shows promise." And added:**

"A robust relationship between PO₃ and ozone concentrations can only be established by factoring in physical processes such as horizontal and vertical transport, dry deposition rates, and background values. In regions with high background ozone concentrations, for example in mountainous areas, even a moderate level of PO₃ can elevate ozone concentration to unhealthy levels. Conversely, if there is a strong correlation between PO₃ and frequent ozone exceedances, such as those observed in the mentioned U.S. cities, it indicates that locally produced ozone through chemical reactions is the primary factor contributing to those events."

**In the summary section, we added:**

"It is important to recognize that PO₃ maps are just one piece of the puzzle when it comes to determining ozone concentrations. Several studies have indicated that accurately representing surface ozone is challenging due to difficulties in representing background ozone, vertical transport, and dry deposition rates (e.g., Zhang et al., 2023; Clifton et al., 2020). Therefore, we advise against directly linking high PO₃ rates from our product to increased unhealthy ozone exposure. However, our product can provide indications as to whether heightened ozone concentrations are associated with rapid local chemistry as opposed to other processes (e.g., meteorology or dry deposition rates). Further investigation using additional tools/data is necessary to gather a full picture of these processes."

Minor Comments:

In the first paragraph the authors make some general statements about the importance of ozone for health, vegetation and climate, but provide no references. This would be a good opportunity to cite the findings from the first phase of TOAR in three key publications, TOAR-Health (Fleming et al., 2018),

TOAR-Vegetation (Mills et al., 2018) and TOAR-Climate (Gaudel et al., 2018).

**Thanks, we now have included them.**

| Response |
| --- |
| **Thanks, we now have included them.** |
| **Modifications** |
| Ozone not only poses significant risks to human health (Fleming et al., 2018) and agricultural productivity (Mills et al., 2018) but also influences the radiation budget, thereby affecting the climate (Gaudel et al., 2018). |

Line 199 SZA is first mentioned here, but it needs to be defined

| Response |
| --- |
| **Added.** |
| **Modifications** |
| … solar zenith angle (SZA)… |

Line 679 "This data has" should be "These data have"

| Response |
| --- |
| **Corrected.** |
| **Modifications** |
| These data have indeed … |

---

## Author Comment (AC2)

The study by Souri et al. uses a combination of a box model, CTM output, satellite data and aircraft data to try and estimate ozone production rates in the lower boundary layer. The aim of the paper definitely sits within the remit of ACP and TOAR-II, but I believe the manuscript needs major corrections (though mainly textual) are required before it can be accepted for publication.

**We thank this reviewer for their constructive comments. Our response is as follows:**

Major comments:

1. So, when I read the title and abstract of the paper, it read as if the satellite data was the main dataset/resource used to general the PO3 maps. However, from a detailed read of the manuscript, lots of other data sources are required to achieve the outcome. For instance, the authors use model output from a CTM to derive boundary layer satellite NO2 and HCHO products. This is fine but you have moved a long way from "using satellite data". However, the bias correction of the TROPOMI HCHO and NO2 using surface column measurements is a good practical step to achieve more robust results. There is then the box-model, which is partial tuned to aircraft observations for some tracers, to evaluate key variables which will go into the final scheme (shown nicely in Figure 2) to derive PO3. Overall, I am happy with the methods used to derive PO3 (especially as the satellite data gives you the high spatial resolution) but I think the actual overarching method of the paper needed rewriting (i.e. instead of putting the emphasis on "using satellite data", I think you should make it clearer that you use "a synergy of data products" to derive high spatial maps of PO3.

**Response**

**We agree with the reviewer that the product is not something that solely relies on satellite radiance. As a matter of fact, even satellite retrievals are derived from a combination of models, auxiliary data (albedo, snow/ice information ,…), and the satellite radiance info. To incorporate this valid point, we have done several adjustments to the text:**

**Modifications**

**We have renamed the title:**

*"Feasibility of robust estimates of ozone production rates using a synergy of satellite observations, ground-based remote sensing, and models"*

**Because we use various models (M2GMI, NCAR TUV, and F0AM) we decided to use a generic name (models) in the title.**

**In the introduction we added:**

*"Inspired by those works, we developed a novel product using TROPOMI observations in conjunction with ground-based remote sensing and atmospheric models to estimate PO₃ and associated errors within the planetary boundary layer (PBL) across the globe."*

**In the summary section:**

*"Providing data-driven and integrated maps of ozone production rates (PO₃) using a synergy of satellite retrievals, ground-based remote sensing, and atmospheric models enabled us to generate the first satellite-informed product of this kind, offering extensive spatial coverage with significant applications in atmospheric chemistry."*

**In a figure caption, we removed a sentence and replaced it with:**

*"These estimates are based on the proposed algorithm integrating TROPOMI, ground-based remote sensing, and atmospheric models, to estimate PO₃ based upon a statistical approach."*

2. In sections 2.2 and 3.1, there are multiple equations but some of the variables are not actually defined, which made it difficult to fully understand the methods without being an expert. So, these sections need to be improved to clearly define and explain what all the variables are in the equations. For instance, on line 161, what are a, b and ε? On line 162, what does i represent? There are also several examples where variables have not been added to the equations (e.g. line 175...I assume these are superscript 2s?). Overall, the method's presentation needs to be improved and discussed more to make it clear to non-experts what you are using the methods for. There are examples below in the Minor Comments supporting this.

**Response**

**Thanks for this comment, we now have improved these sections to better define the equations/methods.**

**Modifications**

**We added more description of the variables in Section 2.2.1:**

*"To achieve an optimal linear fit ($y = ax + b + \varepsilon$) between the paired observations, where a and b are slope and offset to be determined, we follow a Monte-Carlo Chi-squares minimization such that $\chi^2 = \sum \frac{[y - f(x_i, a, b)]^2}{\sigma_y^2 + a^2 \sigma_x^2}$ is minimized. In this equation, $\sigma_y^2$ and $\sigma_x^2$ are the variances of y (TROPOMI) and x (the benchmark, here MAX-DOAS or FTIR), respectively; i is the subscript refers to i-th observation point, and f is the proposed linear fit subject to optimization."*

**In Line 175, we can either describe the variable or the squares of the variable; it is not necessary to describe the math operation in equations; but to increase the clarity we added:**

*"Since there are errors associated with this adjustment resulting from instrument and representation errors, we augment errors of the slope and offset to the total error and label them constant errors ($e_{const}^{\square}$) via:*

$$e_{const}^2 = e_{offset}^2 + e_{slope}^2 \times VCD_{bias-corrected}^2 \qquad (2)$$

*where $e_{offset}^2$ and $e_{slope}^2$ are squares of errors of offset and slope calculated from the linear regression (Eq. 1). Ultimately, the sum of all three errors constitutes the total errors given:*

$$e^2 = e_{const}^2 + \frac{1}{m^2} \sum_{i=1}^{m} e_{random,i}^2 \qquad (3)$$

*where m is the number of samples for a given grid and timeframe and $e_{random}^2$ is squares of random errors."*

**In section 3.1, we added:**

*"To achieve this, we can use LASSO (least absolute shrinkage and selection operator) (Tibshirani, 1996) consider a regression,*

$$Y = X\beta + \alpha + \varepsilon \tag{5}$$

*with response $Y = (y_1, ..., y_n)^T$, $n \times p$ explanatory variables $X$, coefficients $\beta = (\beta_1, ..., \beta_p)^T$, an intercept $\alpha$, and noise variables $\varepsilon = (\varepsilon_1, ..., \varepsilon_n)^T$. n is the number of data points, and p is the number of explanatory variables. We can label the regression model sparse when many of $\beta$ values are zero, and we can label it high dimensional when $p \gg n$. LASSO attempts to select variables such that the following cost function is minimized:*

$$(\hat{\alpha}, \hat{\beta}) = argmin \left\{ \|Y - X\beta - \alpha\|_2 + \lambda \sum_{i=1}^{p} |\beta_i| \right\} \tag{6}$$

*where $\hat{\alpha}$ and $\hat{\beta}$ are optimized intercept and coefficients, $\lambda$ is a non-negative regularization factor subject to tuning, i is the subscript of the i-th explanatory variable, and $\|.\|_2$ is the L2-norm operator."*

Minor Comments:

Line 100: Define NASA and NOAA in the first instances.

| Response |
|---|
| **We defined them.** |
| **Modifications** |
| To study PO$_3$, we use various aircraft observations from several National Aeronautics and Space Administration (NASA) and National Oceanic and Atmospheric Administration (NOAA) atmospheric composition campaigns. |

Line 153: "a fixed additive component that is magnitude-independent", can you provide more detail on what this is and why you use it.

| Response |
|---|
| **This is the offset derived from the comparison of TROPOMI and the ground remote sensing data. A uniform error that exists everywhere in the scene and it does not vary with the VCD magnitudes.** |
| **Modifications** |
| **We added:** |
| To propagate TROPOMI retrieval errors to the PO$_3$ product and to remove potential biases, we assume three origins for errors: i) random errors resulting from instrument noise, ii) a fixed additive component that is magnitude-independent (i.e., a uniform offset persisting over all pixels), and iii) unresolved systematic biases that are multiplicative and irreducible by oversampling. |

Line 163: Can you give an example of what you mean by "benchmark".

| Response |
|---|
| **We added FTIR or MAX-DOAS in the parenthesis.** |
| **Modifications** |
| … and $x$ (the benchmark, here FTIR or MAX-DOAS), respectively. |

Line 154: Can you use the satellite column precision as a representation of random errors?

**Response**

**Random errors are mostly dictated by the random errors in the slant column fit. In fact, AMF do not have significant random errors as most of its inputs rely on models (RTM and CTM) or averaged values (except for the O2-O2 algorithm). Here, by the random errors, we strictly refer to errors coming from the pixel SNR (depending on the scene radiance and the instrument noise) and how strong the absorption lines for NO2 and HCHO molecules are depending on their vertical distributions/magnitudes. Both of these components can be well approximated by the error of the fit in SCD projected onto VCD using AMF. The information about this error varying by pixel to pixel is articulated by the precision error variable coming with the L2 product.**

Line 156: Can you be clearer on the text "Moreover, to mitigate this error, its squares are average over a month".

**Response**

**Thanks we have clarified it. The random noise gets beaten down by 1/sqrt(n), where n is the number of available pixels in a month for a given area.**

**Modifications**

Moreover, we average the squares of random errors over a month to reduce random noise by the squared number of pixels available at the same location.

Line 161/2: Please define what a, b, ε and i are.

**Response**

**We now defined them (mentioned above).**

Line 175: Missing variables in boxes.

**Response**

**Corrected.**

Line 184: What does BRDF stand for?

**Response**

**The bidirectional reflectance distribution function.**

**Modifications**

*"The product has outperformed traditional LER products such as OMI when both were compared to MODIS surface the bidirectional reflectance distribution function (BRDF) results (Tilstra et al., 2024)."*

Section 2.4: Please provide more information on how you use the MERRA2-GMI data to generate the satellite PBL product?

**Response**

**Thanks, we added the equation.**

**Modifications**

To carry out the conversion, we apply the following conversion factor (γ) to the TROPOMI VCDs:

$$\gamma = \frac{\bar{q}_{PBLH}}{\frac{NA}{g \times Mair} \Sigma \, qdp} \qquad (4)$$

where $\bar{q}_{PBLH}$ is the average of the target trace gas mixing ratios in the PBLH, $g$ is the acceleration of the gravity (assumed 9.81 m/s$^2$), $NA$ is the Avogadro constant, $Mair$ is the air molecular weight (assumed 28.96 g/mol), $q$ is the target trace gas mixing ratio at a given altitude, and $dp$ is the thickness of each model vertical grid box in hPa. The denominator in Eq. 4 represents the modeled VCD. We integrate modeled partial VCDs up to top of the atmosphere for HCHO, and up to the tropopause pressure layer for NO$_2$.

Line 217: What do n and p represent?

| Response |
| --- |
| We now have defined them. |
| **Modifications** |
| $n$ is the number of data points, and $p$ is the number of explanatory variables. |

Equation 5 RHS: Should "2" be superscript instead of subscript?

| Response |
| --- |
| No, ‖.‖2 is the notation for the L2-norm operator. We now have clarified it. |
| **Modifications** |
| *… and $\|.\|_2$ is the L2-norm operator.* |

Line 227: Please make it clearer what you mean by "folds".

| Response |
| --- |
| This is a generic term used in cross-validation algorithms. We clarified it. |
| **Modifications** |
| *To optimize this value, we discretize $\lambda$ in 100 values between $10^{-4}$ up to $10^1$, divide the training dataset into 10 folds (i.e., spliting the dataset into equal size segments),* |

Lines 353/354: "Consequently, it is likely that the measurements error resulted in more spread in the comparison". Can you provide some references to support this statement.

| Response |
| --- |
| The random noise associated with NCAR's NO$_2$ and NO measurements are reported to be around 0.05 and 0.01 ppbv for NO$_2$ and NO, respectively. In the log-space, they will be around -1.3 and -2.0. So, the reason that we see a fatter distribution in the comparison over pristine areas is that the uncertainty of the measurements go beyond 100% (blue circles): |

[Figure]

**We think it is important to acknowledge that NO and NO2 measurements in remote regions (or high altitudes) can be highly uncertain (Shah et al., 2023).**

| Modifications |
| --- |
| **We added:** |

*"Consequently, it is likely that the measurements error resulted in more spread in comparison. In particular, Shah et al. (2023) found that these measurements could be contaminated by various reactive nitrogen species in remote regions precluding a robust validation of atmospheric models."*

Line 364: Rephase "not to unrealistic" to "reasonable".

| Response |
| --- |
| **Corrected.** |

Figure 3: NO2 and NO have the same MB, MAB and RMSE. Is this a duplicate of statistics or coincidence? Also, some of the stats legends overlap (e.g. OH), so the presentation needs to be improved here.

| Response |
| --- |
| **We doubled checked the code. The stats are calculated by a function inside the loop iterating over each specie. They are the same within 2 decimal point precision. They are not identical. We have recreated the figure to remove the overlaps.** |

Line 555: One could argue why don't use just use a CTM or regional model to simulate/output PO3 and supporting variables (e.g. NO2 and HCHO). Would you not benefit from using the satellite and aircraft observations to evaluate the model, identify limitations (e.g. emissions, chemical mechanism etc.), undertake sensitivity experiments to resolve the limitations and then provide more robust estimates of PO3 from the model? That way, you are getting estimates of PO3 but also improving the processed based model providing a better understanding of the processes governing PO3?

**Response**

**This is a valid point. It's more physics-based to incorporate satellite observations into a model to optimize the model's prognostic inputs. This allows us to see the chain of adjustments on various physiochemical processes within a process-based framework. Souri et al. 2020 (https://acp.copernicus.org/articles/20/9837/2020/ ) was a pioneering work that adjusted NOx and VOCs emissions simultaneously using multi-sensors to better represent PO3 across East Asia. This was motivated by the importance of the chemical feedback between NOx-VOC and HCHO-NOx. However, as we mentioned in the introduction (the paragraph starting with "While the characterization of ozone regimes …"), it is prohibitively expensive to perform joint inversions globally and for a long-term record. Data-driven approaches, like the one described in our work, provide a shortcut. It will neither replace a constrained chemical transport model capable of providing all physiochemical processes and reaction rates, nor will it be a product to understand ozone chemistry. It is simply an estimate of PO3 maps along with the contribution ones that can provide more detailed information compared to binary maps obtained from FNR. As we mentioned in the summary, our parametrization can be enhanced through using more sophisticated algorithms that are better capable of capturing the non-linear chemistry associated with ozone. An upcoming part of our project will demonstrate the use of deep-neural networks that have been shown to predict PO3 and the derivatives without the need for FNR. We believe these new statistical approaches can provide rapid results for regulators to implement emission mitigation in a timely manner, and to prioritize sub-orbital missions over places where in-situ measurements are absent.**

Line 560: There are a few instances where you term "significant". However, do you actually use a statistical test to support these statements?

**Response**

**No, we unfortunately didn't use a statistical test. So we removed this term whenever we say something might be significantly different than the other term.**

Figure 11: I might have missed this but do you define "CONUS"?

**Response**

**Now defined in the beginning of the paragraph.**

Figure 17: "The data is based on 2019 TROPOMI observations". This does not make sense. You list several variables on Lines 652 only which two are actually from TROPOMI. Please update this.

**Response**

**We have modified this sentence along with any other sentences that may wrongly imply that the estimates were purely derived from satellite radiance.**

**Modifications**

*"These estimates are based on the proposed algorithm integrating TROPOMI, ground-based remote sensing, and atmospheric models, to estimate PO₃ based upon a statistical approach."*

Line 678: I disagree with this statement "satellite-derived product". As to my major comment #1, you use satellite data, aircraft data, MAX-DOAS data, CTM data, box model data and statistical methods to derive PO3 (as depicted in your Figure 2). Therefore, I believe this needs to be reworded and refocussed (e.g. a data-model fusion approach to derive PO3 etc.).

**Response**

**We modified this part. A large fraction of these estimates come from the satellite information so we believe the "satellite-informed" attribute of our product should be highlighted.**

**Modifications**

*"Providing data-driven and integrated maps of ozone production rates (PO₃) using a synergy of satellite retrievals, ground-based remote sensing, and atmospheric models enabled us to generate the first satellite-informed product of this kind, offering extensive spatial coverage with important applications in atmospheric chemistry."*

---

## Author Comment (AC3)

**Main Comments:**

The authors develop a powerful parameterization of ozone production rate using satellite-derived columns of NO2 and HCHO along with modeled photolysis rates. Overall, an excellent, important, manuscript, although it can be difficult to follow at times.

**We thank this reviewer for their constructive comments. Our response is as follows:**

L206: At the beginning of the methods section, add a few sentences explaining how sections 3.1-3.4 fit together.

| **Response** |
| --- |
| **We added a few sentences to describe the sections.** |
| **Modifications** |
| "In this section, we begin by discussing a robust regression model specifically developed for feature selection in the parameterization of $PO_3$. We then describe the training dataset created for this purpose. Following that, we introduce a clustering technique utilized to organize the training data, which enables us to identify the key drivers of $PO_3$ variability. Finally, we provide a comprehensive overview of the $PO_3$ estimates algorithm by integrating data from the TROPOMI retrievals, ground-based remote sensing, and various models." |

L274-L296: How does the clustering described in section 3.3 fit into the rest of the paper? Is it part of the coefficient determination or just an analysis tool. Please explain more clearly.

| **Response** |
| --- |
| **The clustering algorithm was an auxiliary tool to pinpoint the major drivers of $PO_3$ variability as well as to show that a wide range of atmospheric conditions has been covered in our study.** |
| **Modifications** |
| **We modified the section by starting:** |
| "The aim of using a classifier to group the large quantity and types of aircraft data into similar features is to allow us to study the primary contributors to $PO_3$ under different chemical, solar, and meteorological conditions. Additionally, this approach will help us understand the range of atmospheric conditions included in the training dataset." |

L400: How did you end up with 7 distinct classes after your clustering analysis? Was it trial and error based on how the deviations of observations from the centroids of the 11 features looked?

| **Response** |
| --- |
| **While some statistical tools (such as the silhouette metric) can help find the optimum number of classes, we found the number of classes sufficient to explain their distinctive characteristics with respect to solar radiation, FNR, FNP, and altitude. Almost every class has a unique feature, allowing us to explain their differences quickly.** |

L524-543. It appears that you adjust the TROPOMI NO2 and HCHO to remove biases with respect to MAX-DOAS and FTIR observations. How important is this result to your bottom line coefficients for PO3 and wouldn't these biases be regional and subject to change with new versions of TROPOMI data?

| **Response** |
| --- |
| **Thanks for the comment; the coefficients determined for the $PO_3$ parametrization rely only on the training dataset obtained from the observationally-constrained F0AM model. They do not depend on the satellite dataset. What can change that coefficient is the inclusion of a new air** |

**quality campaign or different configurations in F0AM. The statistics we gain from TROPOMI errors can change with new updates to the retrieval algorithm or new benchmarks; however, as far as we know, the present work is the most comprehensive validation of the most recent TROPOMI dataset. It is also important to recognize that some biases associated with these data are systematic and do not change drastically from a specific version of the product to another. For instance, the large underestimation of both HCHO and NO2 columns has been widely recognized in literature based on various subsets of ground-based remote sensing measurements (please see Table 1 in https://acp.copernicus.org/articles/21/18227/2021/).**

**Minor Comments**

L93 Break this paragraph into two with the last paragraph previewing what you are doing in this manuscript.

| **Response** |
| --- |
| **Thanks, we divided them into two pieces.** |

L127: Is there a version number for these recently reprocessed fields?

| **Response** |
| --- |
| **Yes, we added it.** |
| **Modifications** |
| We use the recently reprocessed daily level-2 (L2) TROPOMI tropospheric $NO_2$ and total HCHO columns (v2.4) derived from UV-visible radiances onboard the European Space Agency's (ESA's) Sentinel-5 Precursor (S5P) spacecraft (~328-496 nm). |

L199-200: I don't understand the meanings of the colons within the parentheses. Are these threshold values for the look up tables? If yes, why do the values jump around so much such as 100:50:600?

| **Response** |
| --- |
| **Sorry for the confusion! The first and the last numbers are the boundaries and the middle number is the interval.** |
| **Modifications** |
| **We modified the sentence to:** "This look-up table is based on the calculation of more than 20,064 solar spectra over a wide range of solar zenith angle (SZA) (the range [0, 90] in steps of 5°), altitude (the range [0, 15] in steps of 1 km), overhead total ozone column (the range [100, 600] in steps of 50 DU), and surface UV albedo (the range [0, 1] in steps of 0.2) using NCAR's Tropospheric Ultraviolent and Visible radiation model (TUV v5.2) and cross sections and quantum yields from IUPAC and JPL (Wolfe et al., 2016)." |

L268: Do you really mean equation 3 here?

| **Response** |
| --- |
| **Corrected.** |

Figure 2. Perhaps italicize SZA, ambient temperature, and Pressure as they are dropped?

| **Response** |
| --- |
| **Done.** |

Figure 2: Why is the left column labeled "Input Candidates from Aircraft" when it uses model and satellite data?

**Response**

**The inputs for the parametrization (i.e., training data set) come from the F0AM model constrained by aircraft dataset. For the prediction (the right part of the flowchart) we use satellites and models.**

L325: M2GMI (be sure to define somewhere)

**Response**

**Defined.**

L359-L364: Here or perhaps in section 3.2, Add some background on the role dilution factors play in box model calculations.

**Response**

**Thanks, we had already dedicated few sentences to talk about the role of dilution factor:**
*...As a result, we have simplified the physical loss by employing a first-order dilution rate set to 1/86400 s-1, equivalent to a lifetime of 24 hours. This approach ensures that unconstrained trace gases that take longer to break down do not accumulate over time. Exact knowledge of dilution factors requires knowing molecular and turbulent diffusion, entrainment and detrainment, and deposition rates, all of which are unknown at the micro-scale level of aircraft observations. Nonetheless, studies of Brune et al. (2022) and Souri et al. (2023) showed that HO2, OH, NOx, and HCHO are relatively immune to the choice of the dilution factor, whereas RO2 mixing ratios can depart introducing some biases in PO3 estimates….*

L391-397: The last 3 sentences of the Figure 3 caption contain information that is also in the main body of the article. Perhaps delete. You probably should mention which field campaigns had the most observations and therefore played the largest role in determining the statistics.

**Response**

**Removed.**

Figure 4: I notice you use log(FNR) and log(FNP) here. Could you explain the benefits of this transformation.

**Response**

**Both FNRs and FNP can have extremely large values making it more difficult to put both low and high values in the same plot. Therefore, the use of log() was helpful to have all of them on the same plot with minimal spaces.**

**Modifications**

**We added the following sentence to Figure 4:**
"Both FNR and FNP are scaled using the logarithmic function to enable the simultaneous visualization of low and high values within a single plot."

L449-451. Did ambient T, H2O vapor, pressure and/or SZA add any additional insights? Preview the results here.

**Response**

**H2O is known to influence ozone through O1D+H2O->2OH, and many chemical reactions rely on temperature and pressure. As mentioned in the paper, we can't say if they add new information based on the clustering algorithm. Still, we decided to include them in the LASSO estimate so the L1-regularization could determine if they are helpful at better predicting PO3. The LASSO algorithm didn't consider them for several reasons: i) we think there is a strong correlation between HCHO and temperature, so HCHO data already have temperature information included; ii) SZA and photolysis rates are highly correlated; and iii) H2O has non-linear effect on PO3 due to generation of 2OH molecules (please see https://agupubs.onlinelibrary.wiley.com/doi/full/10.1029/2019GL084486) ; since we did not separate the regression into different humid regions, the LASSO algorithm was unable to use H2O in a linear form. This is an inherent limitation of the LASSO algorithm, and for this reason, we had mentioned in the conclusion that we may need to explore the capabilities of a more sophisticated algorithm (i.e., deep neural network) to consider all non-linearities without having to linearize the problem using various ozone indicators.**

L467-469: Earlier you mention that SZA, pressure, and temperature were dropped. Here you also include H2O vapor.

**Response**

**Thanks for noticing this! We added water vapor to the text.**

L621: Be sure to expand the Benelux acronym the first time it is introduced.

**Response**

**Added.**

Figure 17. You may want to change the order of the contributions so that the third listed contribution in the legend (jNo2) is also the third in the Figure (it is currently the second from the top).

**Response**

**The legend follows the order of the "area" function in MATLAB starting from the bottom to the top part of the charts. So we decided to leave it as is.**

L756-773: The financial support section lists numerous measurements some of which seem to have little relation to this project. Would it be possible to tighten this section up by eliminating data sets that are only peripherally related to this study while adding more information on how particular measurements were important for this study.

**Response**

**While we fully understand the reviewer's concern, it is mandatory for us to include all FTIR and MAX-DOAS contributions in the acknowledgment. It is part of their terms of use. The correction factors derived from these datasets had a significant effect on our results because the slopes were far from one.**

**Grammatical Comments:**

L96: use degrees symbol.

| **Response** |
| :--- |
| Corrected. |

Section 2.4.  …. But how do you convert the VCDS?

| **Response** |
| :--- |
| **Thanks, we added the equation.** |
| **Modifications** |
| To carry out the conversion, we apply the following conversion factor (γ) to the TROPOMI VCDs: |

$$\gamma = \frac{\bar{q}_{PBLH}}{\dfrac{NA}{g \times Mair}\sum qdp} \tag{4}$$

where $\bar{q}_{PBLH}$ is the average of the target trace gas mixing ratios in the PBLH, $g$ is the acceleration of the gravity (assumed 9.81 m/s$^2$), $NA$ is the Avogadro constant, $Mair$ is the air molecular weight (assumed 28.96 g/mol), $q$ is the target trace gas mixing ratio at a given altitude, and $dp$ is the thickness of each model vertical grid box in hPa. The denominator in Eq. 4 represents the modeled VCD. We integrate modeled partial VCDs up to top of the atmosphere for HCHO, and up to the tropopause pressure layer for NO$_2$.

L197: To estimate photolysis rates of JNO2 and JO1d-◊ To estimate the photolysis rates, JNO2 and JO1d), we

| **Response** |
| :--- |
| Corrected. |

L216:  -->  (Tibshirani, 1996).  They consider a regression,

| **Response** |
| :--- |
| Corrected. |

L292: These features include -->  These features are

| **Response** |
| :--- |
| Corrected. |

L317: are based on converted the bias-corrected --> are derived by converting the bias-corrected

| **Response** |
| :--- |
| Corrected. |

Figure 2: Typo.  Should be M2GMI Conversion Factor within the diamond.

| **Response** |
| :--- |
| **Thanks for noticing this! Fixed.** |

L480: more photolysis rates --> higher photolysis rates

| Response |
| --- |
| **Corrected.** |

L487: by random dropping --> by randomly dropping

| Response |
| --- |
| **Corrected.** |

L513: predictor power --> predictive power

| Response |
| --- |
| **Corrected.** |

L596: making NO2 levels -->  meaning NO2 levels

| Response |
| --- |
| **Corrected.** |

L686: maps of within the PBL --> PBL maps

| Response |
| --- |
| **We fixed this sentence.** |

| Modifications |
| --- |
| "In this study, we generated PO$_3$ maps within the planetary boundary layer (PBL), constrained by bias-corrected TROPOspheric Monitoring Instrument (TROPOMI) observations, using a piecewise regularized regression model." |

---

## Author Comment (AC4)

During the revision of the manuscript, we identified an error in the presentation of the error maps for PO₃ estimates. The errors had been incorrectly represented using daily TROPOMI data instead of the intended monthly data. As a result, the estimated errors for the monthly-averaged maps were significantly overestimated. The updated figure shows that the errors in PO₃ estimates are now presented as being within 10-20% over polluted regions, which is an improvement from the previous 40-60% range. Additionally, in remote regions, the errors can exceed 30% instead of the prior estimate of 100%. Although this update was unintended, it reinforces our claim regarding the feasibility of PO₃ estimates with even a smaller margin of error.

[Figure]

**Figure 18.** The influence of the satellite errors on PO₃ estimates (absolute and relative) over four major regions tackled in this work. The errors are based on monthly-averaged TROPOMI errors. The errors tend to be mild over polluted regions (10-20%) but they can exceed above 50% over pristine ones.